## Registered report

physiology/neuroscience/psychology

cortical thickness, brain and behaviour, social cognition, cognitive abilities, face cognition, core brain network of face cognition

**Author for correspondence:**
Kristina Meyer
e-mail: kristina.meyer@uni-oldenburg.de

# Are global and specific interindividual differences in cortical thickness associated with facets of cognitive abilities, including face cognition?

Kristina Meyer[1], Benjamín Garzón[2], Martin Lövdén[2] and Andrea Hildebrandt[1]

[1]Department of Psychology, Carl von Ossietzky Universität Oldenburg, Oldenburg, Germany
[2]Aging Research Center, NVS Department, Karolinska Institutet and Stockholm University, Tomtebodavägen 18A, 17165 Stockholm, Sweden

 KM, 0000-0003-0232-0211

Face cognition (FC) is a specific ability that cannot be fully explained by general cognitive functions. Cortical thickness (CT) is a neural correlate of performance and learning. In this registered report, we used data from the Human Connectome Project (HCP) to investigate the relationship between CT in the core brain network of FC and performance on a psychometric task battery, including tasks with facial content. Using structural equation modelling (SEM), we tested the existence of face-specific interindividual differences at behavioural and neural levels. The measurement models include general and face-specific factors of performance and CT. There was no face-specificity in CT in functionally localized areas. In *post hoc* analyses, we compared the preregistered, small regions of interest (ROIs) to larger, non-individualized ROIs and identified a face-specific CT factor when large ROIs were considered. We show that this was probably due to low reliability of CT in the functional localization (intra-class correlation coefficients (ICC) between 0.72 and 0.85). Furthermore, general cognitive ability, but not face-specific performance, could be predicted by latent factors of CT with a small effect size. In conclusion, for the core brain network of FC, we provide exploratory evidence (in need of cross-validation) that areas of the cortex sharing a functional purpose did also share morphological properties as measured by CT.

# 1. Introduction

Throughout human history, people have depended on the cooperation with others in order to survive and to strive. Interpreting the faces of those around us is a crucial prerequisite for mastering this social inter-dependence, since faces provide unique information of paramount importance to social interaction. Evolutionary psychologists have argued that facial expression recognition aided survival in a prehistoric society [1] and that choice of an appropriate partner is still influenced by facial features [2,3]. From an evolutionary perspective, it would therefore not be surprising if the human central nervous system had developed a highly specialized systems for processing faces. The most conclusive evidence indicating that face cognition is indeed a specific ability arises from observations of prosopagnosia [4,5]. The specific impairment of face processing abilities in prosopagnosic patients, accompanied by unimpaired ability to process other visual objects, leads to the assumption that there are specific neural structures enabling processing of face-related information.

Work in the psychometric tradition has also reported evidence of face-specific individual differences in samples of normal healthy adults. Wilhelm and colleagues [6] systematically assessed the speed (reaction time in easy tasks) and accuracy (proportion correct in difficult tasks) of performance of young adults (two samples; $N = 151$, $N = 209$) on face and object cognition tasks. By means of structural equation modelling of the data acquired from perception and memory tasks, Wilhelm and colleagues [6] established content-specific (faces versus non-faces) as well as operation-specific (perception and memory) latent factors of processing accuracy. The factors of face perception accuracy and face memory accuracy captured systematic and substantial individual differences beyond object cognition and general cognitive abilities. However, one latent factor was sufficient to account for individual differences in reaction times in all easy speed tasks, independently of whether faces, houses, letters, numbers or abstract figures have been processed. In further studies, the differentiated ability factors for accuracy and less differentiated ones for speed have been additionally shown to be invariant across the adult lifespan [7–9].

## 1.1. The neuroanatomy of face cognition

Early functional brain imaging evidence of face-selective regional brain responses gave rise to the modular theory of face processing (see [10] for a review). However, strict modularity could not be held up for long, since brain regions that are active during face processing seem to respond in varying degrees also to other stimuli (e.g. [11,12]). For instance, the fusiform face area (FFA), although it usually revealed particularly strong selectivity for faces, was later associated with other objects of expertise [13]. There is now a shift of view toward a network perspective. We know from decades of extensive research on the nervous system that, for the sake of neural efficiency, there are only few specialized cells fulfilling only a single function in the neocortex. Instead, most parts of the brain seem to be adaptively recruited in multiple situations, hence serving more than one function [11,14]. It is the interplay between distributed cells or cell clusters which accounts for specificity: in the context of specific tasks, a cell cluster might interact quickly with different other sections of the brain, momentarily forming a specialized network. Thus, in line with modern neuroanatomical and neurofunctional views, brain networks rather than modular brain areas are now considered to be involved in specific abilities [15–17]. Several face cognition abilities have been associated with a set of brain areas that seem to interact to varying degrees in different persons and situations. Within the framework summarized as the model of distributed neural systems of face processing, the identified network has been divided into a core and an extended network of face processing [18–20]. The FFA holds a central position within the core network [10,21], along with the occipital face area (OFA) [22] and the posterior superior temporal sulcus (pSTS) [23]. Haxby & Gobbini [20] reviewed a number of studies, which identified the structures belonging to the core network in the fusiform gyrus to be especially active when invariant facial features (face identity) are processed. Further, the pSTS seems to be active during tasks challenging the processing of changeable facial features, such as facial expressions, gaze direction and generally animated facial movements [24–26].

## 1.2. Brain structure and cognitive abilities

Performance in various behavioural tasks has been associated with the volume, thickness and extension of the surface area of task-relevant brain regions [27–33]. Such structure–function associations may have several

origins. More developed brain structure and associated better performance may have genetic origin. However, skill acquisitions during early and adult development may also shape brain structure [34].

Most widely used measures of brain structure include cortical thickness (CT) and voxel-based morphometry (VBM) methods. CT is a surface-based approach (SBA; [35]) where the distance between the pial and the white surface is measured and averaged across a number of vertices. When VBM measures are used [36], the brain is segmented into voxels. Each voxel is then, with a limited degree of certainty, categorized as either grey matter, white matter or cerebrospinal fluid. To measure local cortex volume using VBM measures, the voxels containing grey matter in a certain region of interest (ROI) are counted.

In search of the neural underpinnings of cognitive abilities, several studies have investigated measures of regional grey matter volume in different regions of the brain, aiming to identify reliable neural correlates of intelligence. For example, based on a review of functional and structural brain imaging studies, Jung & Haier [37] suggested the influential *parieto-frontal integration theory* of intelligence (P-FIT). A recent meta-analysis shows results that are largely compatible with the established P-FIT model, but extend it in several ways, including also correlates of cognitive ability in areas of lateral and medial frontal, temporal and occipital cortex and additionally highlighting the importance of neural structures in the posterior cingulate cortex and subcortical structures [38].

The study of structural characteristics within the core regions of the face cognition brain network and related performance outcomes have received less attention. However, individual differences in cortical thickness of the right posterior FFA have been shown by McGugin and colleagues [39] to go along with individual differences in the performance of processing faces and cars. In this study, participants ($n = 27$) completed psychometric task batteries encompassing stimuli of living (such as faces) and non-living (such as cars) objects. Functional and structural brain scans were also acquired, and individual, functionally defined, ROIs of participants' FFAs, segregated into posterior and anterior parts, were used to extract CT of the areas. Results indicated that CT in the FFA predicted both the performance on tasks using faces and living objects and on tasks using other objects of expertise. Especially, there was a relationship between CT in the posterior FFA and face cognition performance. Notably, the authors point out that CT in any part of the FFA was selectively associated with faces. Interestingly, however, while the relationship between CT in the FFA and behavioural task performance was positive for vehicles, indicating that a thicker cortex was associated with better performance, it was negative for faces, indicating that a thicker cortex was associated with diminished performance. McGugin and colleagues [39] interpret this distinction in terms of developmental account of brain learning. Possibly, the time between the acquisition of a new skill and the measurement of CT in the respective brain areas might play a crucial role in explaining these seemingly contradictory results. The authors discuss that skills for evaluating faces were acquired much earlier in life, while car expertise is acquired during adulthood, and mechanisms of plasticity may be different in early as compared to later development.

## 1.3. Aims of the study

We aimed to investigate whether individual differences in specific face cognition abilities are associated with individual differences in the brain structure of these respective areas. As measures of brain structure we focused on estimates of cortical thickness (see *Methods* section, second paragraph). We applied structural equation modelling (SEM) to performance data and structural MR scans [40].

We first aimed to capture specific variability in processing accuracy that is shared by performance estimates in face tasks only, above their shared variance of any cognitive task using non-face stimuli. Next, we investigated face-specific variability in CT by analysing areas of the core face cognition network [20] as compared with whole brain CT. Finally, the measurement models of performance accuracy on the one side and CT at the other side were related by means of a structural model, aiming to test specific relationships between CT and cognitive performance. This approach was deemed advantageous because it specifically distils the interindividual differences that are specific to face processing and to the cortical thickness of the brain regions thought to be involved in face processing.

More specifically, the first model sequence encompassed two successively estimated measurement models of cognitive performance accuracy. The modelling aimed at testing the specificity of individual differences in performance accuracy measured by tasks using faces as compared with other stimuli. In the first model, a single general factor of cognitive ability was established, indicated by all tasks used to capture abilities independently of stimulus specificity (see figure 3 for a depiction of the model and

the *Methods* section for a description of the tasks). Nested under this factor and orthogonal to it, in a second model, a specific factor of face cognition performance was added. We expected that face cognition ability, indicated by tasks with facial content, would account for systematic variance above the general factor of cognitive ability, thus confirming the specificity of individual differences in face cognition.

Importantly, although testing the specificity of face processing accuracy may have provided an opportunity to replicate previous findings, the inferential test of this factorial specificity wasn't framed as an *a priori* hypothesis for the present work. The reason is that because we did not design and evaluate the psychometric measures ourselves; the model was piloted in a small sub-sample we retrieved from an open-access database, the Human Connectome Project (HCP) [41], including 1203 persons. So far, we are not aware of any other latent variable analyses of this behavioural data with respect to the face tasks. In order to ensure that the measures had acceptable psychometric quality (which was deemed a prerequisite of establishing face specificity), we tested the measurement model describing individual differences in behavioural data prior to preregistration (see §3.2 *Preliminary analyses of the behavioural performance data*). For this purpose, we used a sub-sample of $N = 200$ subjects. This sub-sample was then excluded from the final sample.

In a similar way, we tested two measurement models of the CT data. In a first step, we established a one-factorial model assuming that one latent variable sufficiently to explained the variability in CT measures across different regions of the cortex. In a second step, we tested whether including a further factor indicated by CT measured in the core regions of face cognition (the FFA, the OFA and the pSTS; see [20]) substantially increased the model fit. In analogy to the performance model, the face-network-specific CT factor was nested under the general factor of CT. Using this approach, we accounted for general inter-individual variability in CT and tested whether the variance that is incrementally explained by the nested face-specific factor could be attributed to the relatedness of the three areas within the core face processing brain network.

Thus, the general factor of CT encompassed a representative and widespread number of brain regions across the entire cortex. These should include brain regions which have been associated with general intelligence in available systematic reviews [37,38], but should not be restricted to those. The rationale for not limiting the regions to those established for intelligence was that through such a limitation the regions encompassed in the general factor of CT seemed liable to be functionally too similar, so the potential incremental variance that could be explained by the core regions of face processing wouldn't have been attributable to face-specificity (see §3.3.3.1 *ROIs to indicate the general factor of CT* for a detailed explanation of inclusion criteria for areas to be subsumed under the general CT factor). If there is indeed common variance in CT across the face-selective regions of the cortex as compared with general CT (hypothesis 1), we expected the latent factor of CT measured in the core regions of the face processing network to be statistically identified. Furthermore, the addition of the nested factor to the model was expected to substantially increase model fit.

Finally, the two measurement models were related to each other in a structural model. We predicted specific relationships between CT factors and cognitive functions (figure 3), proposing that the general cognitive performance factor would be most strongly related with the general CT factor (hypothesis 2). Furthermore, we expected the CT in the core network of face cognition to predict individual differences in performance of face-specific tasks (hypothesis 3). The test of hypothesis 2 was deemed a prerequisite for hypothesis 3. Assuming an interconnectedness of brain structures, instead of postulating a strong dichotomy, we expected the general CT to also predict individual differences in face cognition performance, but to a lower degree as compared with general cognitive performance (hypothesis 4).

## 2. Methods

We used behavioural and imaging data from the Human Connectome Project (HCP) ([41]; see http://www.humanconnectome.org/ for detailed information and access to all the downloadable data). The HCP open-access project provides a large set of high-quality, state-of-the-art imaging data, including measures of structural as well as task-evoked and resting-state functional MRI. The subjects underwent a variety of different other procedures of data collection, such as genotypization from blood, accompanied by a selection of other physical measures. Crucially for our study aim, the HCP project encompasses behavioural performance data in a large variety of psychometric tasks, both performed inside and outside of the scanner.

SBA and VBM measures are the two most prominent approaches for investigating neural structures (see §1.2). Although VBM measures have their advantages and are widely used, there are certain methodological shortcomings of these techniques [42–44]. In direct comparison, CT, an SBA measure, was found to be more reproducible than VBM [45]. Furthermore, while the anatomical thickness of the cortex is directly represented in CT, the interpretation of VBM measures often remain much less clear [34]. The key measure of VBM techniques, which is a voxel's probability to belong to one of three tissue categories, does not directly and logically represent brain morphology. In the light of these criticisms with respect to VBM and considering that local changes in CT have been used in many modern studies on neural plasticity [46,47], particularly in studies relating face processing abilities to underlying neural structures [39], we applied CT rather than VBM as a measures of grey matter volume.

## 2.1. Sample

The HCP sample currently consists of a total of 1203 young adults. For the present work we chose to include a sub-sample of 1113 persons who completed the 3T MRI protocols. Of these 1113 subjects, 200 were randomly selected for the above-mentioned psychometric evaluation of the tasks (see results below). These 200 subjects were excluded from all further analyses. Among the remaining 913 subjects, 854 have completed both task-evoked fMRI protocols used for localizing the face network (see below). Thus, our final sample contains 854 participants (mean age = 28.74, s.d. = 3.72). This sub-sample comprises 464 females and 178 twin pairs, 113 of which are monozygotic twins.

## 2.2. *A priori* power analysis

An *a priori* Monte Carlo simulation was conducted in M*Plus* v. 7 [48,49] to determine the available statistical power for detecting the hypothesized effects within the postulated models given the sample size of 854 subjects. The models were specified in M*Plus*. To address hypothesis 1, we estimated the power to detect substantial factor loadings on the face-specific CT factor. Due to the novelty of the approach to address face-specificity of CT, we could only rely on very few published effect size estimates for comparable factor loadings. Therefore, we chose not to confine ourselves to only considering studies that observe CT. Instead, we decided that the best possible approximation (given the restricted availability of research applying SEM to CT data) of our research question is to mind studies that used other, comparable brain measures to link mental abilities to neurobiological substrates. For example, Kievit *et al.* [40] applied SEM to fractional anisotropy (FA) data acquired by diffusion MRI, a variable that represents the neural fibre integrity and is therefore considered to be a biological substrate of mental speed. Kievit and colleagues [40] reported loadings on a general factor of FA of around 0.60. Papenberg and colleagues [50] furthermore reported a lateral correlation of 0.60 between grey matter volumes (quantified by using a VBM approach, in contrast to our study) in the lateral frontal cortex and the visual cortex. The standardized factor loadings on the general CT factor were thus set to 0.60. To account for a publication bias in previously reported effect sizes, we additionally investigated the change in power when a range of ±0.10 was drawn around the effect sizes—factor loadings in this case. Because loadings on nested factors are generally lower, the standardized factor loadings on the nested face factor were postulated 0.30. In this case, we do not apply a range of effect sizes, since 0.30 is the lowest, most conservative effect size, which (given the number of indicators loading into the nested factor) will still allow to identify the nested factor. Weaker loadings would arguably lead to potential non-identification of the nested factor. The results of the Monte Carlo simulation showed that the power of inferentially identifying all loadings (even at the lowest boundary of the range for the loadings on the general factor) was at least 0.94 or higher given the available sample size.

To determine the power of detecting the effects postulated by hypotheses 2 and 3, we simulated the structural model combining behavioural and anatomical measures (see figure 3 for a depiction of the model and §1.3 *Aims of the study* for a description of the hypotheses). To this end, the measurement models of accuracy and CT (each encompassing a general factor and a nested, face-specific factor) both entered the structural model exactly as they are described below in the respective paragraphs of the *Analysis and results* section. In contrast to the measurement model of CT, the numerical value of the parameters in the measurement model of behavioural performance was not specified on the basis of *a priori* assumptions, but were derived from the preliminary analyses testing the psychometric quality of the behavioural indicators (see above). For the regression coefficients of performance accuracy on general and face-specific CT measures we assumed differences in their

strength, with the general performance factor on the general CT factor having the highest regression weight of $\beta = 0.40$. The parameter of interest for testing hypothesis 2 is the regression coefficient of the face-specific performance factor on the face-specific CT factor. With an *a priori* assumed regression weight of $\beta = 0.35$, which represents a small effect size according to Cohen [51], given the sample size of $N = 854$ this regression coefficient can be detected with a power of 0.88. The parameters of interest for testing hypothesis 3 are the regression coefficients of the general cognitive performance factor on the face-specific CT and the one of the face-specific performance factor on general CT. These parameters will be compared with the regression coefficient estimated for the face-specific behavioural and CT factors. We expect the crossed regression weights to be weaker than the face-specific regression weight. Thus, in the simulation study we assumed them to be small effects, with a $\beta = 0.20$. Given the sample of $N = 854$, the power simulation revealed a power of 0.82 for detecting the regression of the face-specific performance factor onto the general CT factor and 0.68 for detecting the regression of the general performance factor onto the face-specific CT factor (see figure 3 for a visualization of the model).

## 2.3. Tasks and procedure

### 2.3.1. Outside-scanner psychometric tasks

In the HCP project, a variety of psychometric tasks was administered in separate meetings outside the scanner sessions. To derive performance indicators of general cognitive ability, we used a set of tasks that measured different facets of intelligence: one task measuring working memory (WM), two tasks measuring reasoning ability with figural and spatial content (Gff1, Gff2) and two further reasoning tasks with verbal content (Gfv1, Gfv2; see descriptions below). To derive performance indicators of face cognition ability, we used a facial emotion recognition task (ER) performed outside the scanner, alongside an inside-scanner working memory task (FWM) with facial stimuli.

#### 2.3.1.1. Working memory (WM)

In the working memory task taken from the NIH toolbox [52,53] (http://www.nihtoolbox.org), participants were provided with a sequences of orally and visually presented stimuli, including animals and other objects. After stimulus presentation participants were requested to recall the increasingly long sequences of presented stimuli while sorting them according to their size.

#### 2.3.1.2. Raven matrices (Gff1)

The Gf indicator was derived from a short battery of Raven's matrices [54]. Here, 24 items and three bonus items were administered, arranged in order of increasing difficulty. The items consisted of a three by three matrix of symbols, with one symbol missing at the bottom right. Out of eight given options, participants indicated the symbol that logically completed the matrix. If a subject could not solve five consecutive items, the task was terminated.

#### 2.3.1.3. Line orientation (Gff2)

In this task, participants were shown on the screen two lines of different colour (blue, red) and different spatial orientations for both lines. The blue line had to be rotated by using keyboard buttons until its orientation matched the orientation of the red line. The relative line positions on the screen varied, but the distance between the centres of the lines was invariant across the trials. Whereas the length of the red line did not alter, the blue line could be shorter or longer. There were 24 trials in this task.

#### 2.3.1.4. Oral reading recognition (Gfv1)

In this verbal task taken from the NIH Toolbox [52,53], participants read out loud a sequence of single words presented on the screen. The correct pronunciation and fluency (i.e. whether subjects stumbled) were evaluated.

#### 2.3.1.5. Vocabulary comprehension (Gfv2)

In the vocabulary comprehension task from the NIH Toolbox [52,53], participants were presented with an auditory stimulus, consisting of a spoken word. Thereafter, four pictures were displayed on a screen. Participants were asked to select the picture that matched the word they had just heard.

### 2.3.1.6. Emotion recognition (ER)

An outside-scanner task named the Penn Emotion Recognition task [55] will serve as a proxy for measuring face cognition ability (see [56], who reported face cognition and emotion recognition to be highly related abilities), together with the performance measured on the two localizer tasks described below using faces. Here, a total of 40 faces were presented one at a time and participants were asked to indicate the facial expression which could be one of five choices: happy, angry, sad, scared or neutral.

### 2.3.1.7. Recognition memory of faces (FRec)

After the inside-scanner working memory task using faces (see §2.3.2.1), participants were again presented the faces they had previously seen during the inside-scanner task and had to indicate whether the face was a new face or a familiar face they had encountered in the inside-scanner task. Because the task was conducted after both tfMRI scanning sessions, the results of the two sessions (held on different days) are considered separately. Per session, 24 faces were presented in this task, amounting to a total of 48 faces, 50% of which were new to the participants. Faces were presented intermittently with places.

## 2.3.2. Functional localizer tasks

Functional localizer tasks serve the purpose of identifying brain areas that show selectively increased responses in reaction to the given stimuli. Therefore, they are performed within the MRI scanner in the same session where the acquisition of (typically T1-weighted) images is conducted. There were two separate scanner sessions per participant in the HCP project, mostly completed within two consecutive days. From one session to the next, the scanning direction (right to left or left to right) was altered. Task-evoked fMRI was distributed over both scanner sessions in the sense that the first half of blocks from each task was conducted in the first session, the second half in the second session.

### 2.3.2.1. Working memory task with faces and objects (FWM)

The first localizer task with relevance for the present work was a working memory task with faces and other objects (tools, places and body parts), which consisted of a 0-back and a 2-back block of trials. In the 0-back condition with low workload, one single stimulus was presented at the beginning of the blocked trial sequence. Participants indicated whether each of the presented stimuli in a following sequence matched the stimulus presented at the beginning of the block. In the 2-back condition, participants judged whether a presented face or object matched the one they saw two steps back. The task was split into two runs, which were conducted during the two separate scanner sessions. Per run, two blocks of 10 trials per stimulus category were completed, amounting in a total of 40 trials per stimulus category.

### 2.3.2.2. Emotion processing task (FER; task was not used for any of the indicators in figures 1 and 3)

The second relevant localizer was an emotion matching task. Here, participants decided which one out of two faces presented at the bottom of the screen matched the face at the top of the screen. The faces showed either a fearful or an angry expression. In a control condition, shapes were used instead of faces. Again, the task was distributed over two runs, each containing three blocks of six trials per stimulus type, amounting to a total of 36 trials per stimulus category. We expected that contrasting the face-images in both localizer tasks with non-face images would allow to localize the FFA and OFA, whereas contrasting images showing emotional facial expressions versus abstract shapes was expected to allow us to localize the pSTS.

## 2.4. Demographic variables

Age, gender, handedness and family status were included in our analyses as control variables. Age and gender were acquired by self-report, together with a number of additional demographic variables.

Handedness was assessed by using the Edinburgh Handedness questionnaire [57]. Family status and relations between participants were assessed with the help of the Missouri Family registry. Participants were asked to confirm the registry entries or indicate the relatedness to other study participants in self-reports. Whenever feasible, the self-report and registry information were complemented by means of genotypization in those subjects whose blood had been sampled.

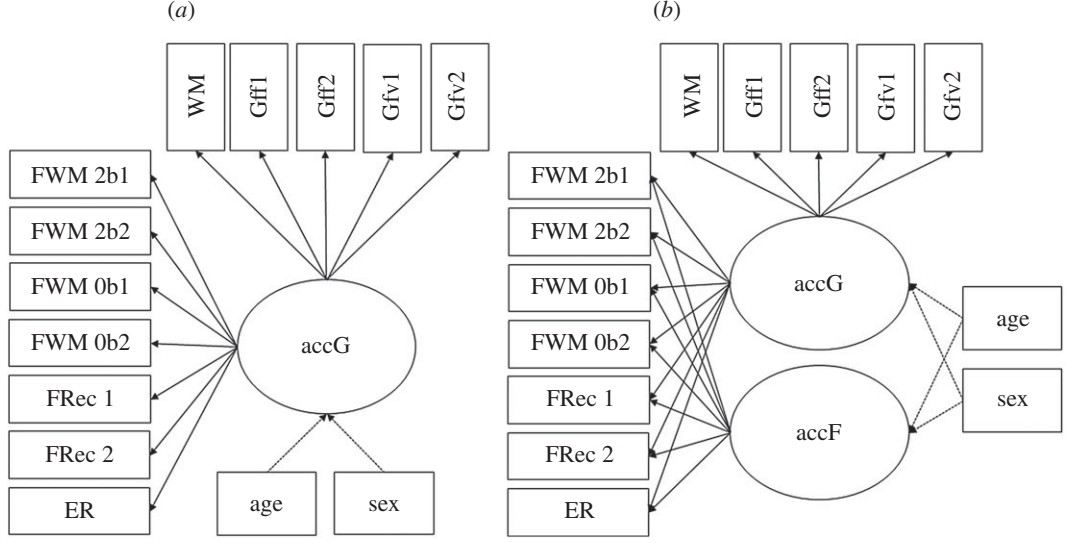

**Figure 1.** (*a,b*) Postulated measurement models for the behavioural data. accG—general factor of performance accuracy; accF—nested factor of performance accuracy in face-related tasks. Indicated by: WM—working memory; Gff1—figural task, progressive matrices; Gff2—figural task, spatial line orientation; Gfv1—verbal task, oral reading recognition; Gfv2—verbal task, vocabulary comprehension; FWM—working memory task with facial content in a 2-back and a 0-back condition; FRec—recognition memory of faces from the inside-scanner working memory task; ER—facial emotion recognition. Age and sex are controlled for by establishing them as covariates (dotted lines indicate covariate influence on the latent factors that will be expected).

## 2.5. Image acquisition

All subjects were scanned using a customized Siemens 3T 'Connectome Skyra' with a 32-channel Siemens head coil. Customized hardware and scanning protocols were applied (see details in WU-Minn HCP 1200 Subjects Data Release Reference Manual, 2017; and appendix 1 of the WU-Minn HCP manual, available at https://www.humanconnectome.org/study/hcp-young-adult/document/1200-subjects-data-release).

Structural scans included T1w and T2w scans. T1w structural scans encompassed a TR of 2400 ms, a TE of 2.14 ms, a TI of 1000 ms, an 8° flip angle and a BW of 210 Hz/Px. The scans were acquired in 7:40 min, the voxel size was 0.7 mm. T2w structural scans were acquired using a TR of 3200 ms, a TE of 565 ms, a variable flip angle and a BW of 744 Hz/Px. Acquisition time was 8:24 min, voxel size was 0.7 mm. Functional scans were collected using a multi-band multi-slice Gradient-echo EPI that underwent improvement and refinement over a number of optimization studies [58–60].

The working memory task was completed in two runs on separate days with 405 frames per run, resulting in a duration of 5.01 min per run. Accordingly, the emotion processing task was also completed in two runs of 232 frames and 2.56 min of duration each.

# 3. Analysis and results

## 3.1. Handling the behavioural data

Statistical data analyses of behavioural data were conducted in R (v. 3.2.3) [61]. To unify the different scales of the acquired task data all variables were z-transformed before entering further analyses. For structural equation modelling (SEM) we used M*Plus* v. 7 [48,49]. The dependent variables observed on a behavioural level comprised accuracy data in both inside-scanner and outside-scanner tasks, reflected in the number of correct trials in each task. Because no psychometric modelling of the above-described tasks is currently available in the literature, we aimed to test the psychometric quality and structure of the available performance indicators prior to preregistration. This was necessary because the behavioural indicators may show ceiling effects or non-normal distributions. Should the behavioural model not be identifiable with the available performance indicators, the main hypotheses for the present work could not be addressed. Thus, in order to ensure that the theoretically derived model of face-specific individual differences beyond general cognitive

functioning was identifiable on the basis of available tasks, we drew a sub-sample of 200 randomly selected persons from the pool of the available participants in the HCP project (see sample description in §2.1). Using the performance data of these 200 persons, we investigated the psychometric properties of the indicators and the model structure prior to preregistration. These analyses did not aim for hypothesis testing, but belonged to an *a priori* data handling procedure. Results are reported in the following paragraph (see also figure 3 and figure 1). The same model was cross-validated in the remaining 854 subjects after preregistration.

## 3.2. Preliminary analyses of the behavioural performance data

First, we inspected uni- and bivariate distributions of all relevant performance indicators (see *Tasks and procedure* in the *Methods* section above). Second, the face-related performance indicators[1] were evaluated with respect to their psychometric properties: we estimated reliability in terms of McDonald's Omega [62] and calculated task difficulties and discriminative power per item (for average behavioural performance in all tasks, see electronic supplementary material, supplement 1[2]). Following psychometric investigations the inside-scanner emotion processing task had to be excluded as indicator of face cognition performance and was only used as localizer. The reason for exclusion was that ceiling effects in this very easy facial emotion matching task compromised the usefulness of this accuracy data for the behavioural model. Ceiling effects were also detected in the outside-scanner emotion recognition (ER) task. However, because the task was of high theoretical interest, we decided against omitting it and instead excluded the easiest conditions, happy and neutral faces, from further analyses (see table 1 for an overview of standardized factor loadings within the measurement models of cognitive performance accuracy). The same trials from the ER task were also included in the cross-validation of the behavioural measurement model estimated in the final sample again after preregistration. In a first measurement model (see figure 1*a*), solely a general factor of cognitive ability accG was estimated. accG was indicated by the number of correct responses in all behavioural tasks. The fit of the one-factorial model was not satisfactory: $\chi^2_{54} = 112.98$, CFI = 0.87, RMSEA = 0.07, SRMR = 0.07. In a second measurement model, additionally to the general factor accG, a nested factor of face-specific cognitive performance accF was added to the model (see figure 1*b*). The factor accF was indicated only be the tasks with facial content. The loadings of the face-specific working memory and recognition memory indicators were significant in the analysed sub-sample, indicating systematic individual differences in face-specific task performance that was not accounted for by the general accuracy factor (see table 1 for standardized factor loading estimates). To provide evidence that the face-specific task performance factor was necessary for explaining the observed data, the variance of the factor was freely estimated and the latent factors were scaled by a reference indicator. In the analysed sub-sample, the nested factor of face-specific performance was identified, $\sigma^2_{accF} = 0.008$, s.e. = 0.003, $p < 0.01$. Furthermore, the inclusion of the face-specific factor clearly improved model fit: $\chi^2_{47} = 66.57$, CFI = 0.96, RMSEA = 0.05, SRMR = 0.05. Some of the indicators included repeated measurements of the same tasks. The inside-scanner working memory task (both 0-back and 2-back conditions) and the recognition memory task were performed during or after two separate scanner sessions on subsequent days. Therefore, we added a third model where the loadings of two indicators representing the same task at two occasions were constrained to equality. In the hitherto analysed sub-sample, the additional constraints did not cause any major deterioration to the model fit: $\chi^2_{49} = 69.15$, CFI = 0.95, RMSEA = 0.05, SRMR = 0.06.

## 3.3. MRI data processing and analysis

Data preprocessing and acquisition of brain measures (cortical thickness and functional contrasts) are encompassed in standardized HCP processing pipelines. Preprocessed and analysed data are directly provided in the HCP repository, alongside the analysis pipelines themselves. Preprocessing includes spatial artefact/distortion removal, surface generation, cross-modal registration, and alignment to standard space. Structural and functional MRI preprocessing makes use of five basic pipelines, incorporating procedures from freely available software packages of FSL [63], FreeSurfer [35] and Connectome Workbench [64]. An overview of the pipelines with a description of their respective

---

[1]In the accepted protocol, we stated to provide McDonald's Omega for 'each indicator'. See §3.5, *Deviations from protocol* for more details.

[2]In the accepted protocol, we used the term 'appendix', which was corrected to 'supplement' throughout the entire article.

**Table 1.** Standardized factor loadings estimated in the behavioural measurement model including $N = 200$ participants. accG and accF—latent variables of performance accuracy in general and face-specific behavioural tasks. Indicated by psychometric tasks: WM—working memory; Gff1—figural task, progressive matrices; Gff2—figural task, spatial line orientation; Gfv1—verbal task, oral reading recognition; Gfv2—verbal task, vocabulary comprehension; FWM—working memory task with facial content in a 2-back and a 0-back condition; FRec—recognition memory of faces from the inside-scanner working memory task; ER—facial emotion recognition.

| model | WM | Gff1 | Gff2 | Gfv1 | Gfv2 | FWM 2b1 | FWM 2b2 | FWM 0b1 | FWM 0b2 | FRec 1 | FRec 2 | ER |
|---|---|---|---|---|---|---|---|---|---|---|---|---|
| accG | 0.42 | 0.68 | 0.60 | 0.81 | 0.76 | 0.42 | 0.27 | 0.36 | 0.31 | 0.21 | 0.29 | 0.24 |
| accG | 0.41 | 0.68 | 0.60 | 0.83 | 0.77 | 0.36 | 0.21 | 0.31 | 0.29 | 0.18 | 0.27 | 0.22 |
| accF | — | — | — | — | — | 0.60 | 0.48 | 0.38 | 0.15 | 0.25 | 0.18 | 0.11 |

major aims is given in the HCP user manual ( pp. 113–117; see also [65]). Details of the preprocessing pipelines can be found in Glasser *et al.* [66]. Importantly, the PostFreeSurfer pipeline performs individual surface registration using cortical folding features and the multi-modal surface matching algorithm (MSM, [67]). In the next step, the MSM registration is complemented by multivariate registration (MSMALL pipeline) on the basis of cortical folding, myelin maps and resting state fMRI correlations. This multivariate registration performs better in large parts of the brain, particularly related with higher cognitive function [67–69]. The HCP task fMRI processing relies on the FEAT utility built in to FSL [70].

For our purpose, we used the functional MSMall-registered scans and cortical thickness maps registered onto the 32 k Conte69 mesh [66], smoothed with surface- and parcel-constrained smoothing of 2 mm full-width half maximum (FWHM). MR data analysis was performed using customized scripts based on the command line tool of Connectome Workbench [64], a software package specifically designed to deal with the HCP data (custom scripts are available via OSF: https://osf.io/p7c8z/). Average CT values extracted from the ROIs were then exported to R for further analyses.

### 3.3.1. Dealing with lateralization

Importantly, we accounted for lateralization of brain areas of face cognition. Evidence suggests that at least a quantitative difference is present between left-sided and right-sided core regions of face processing, with the right-hemispheric regions consistently showing stronger activation in contrast with face versus non-face stimuli and larger sizes of activated voxel clusters (see [71] for review). Some studies furthermore suggest qualitative differences. For instance, Meng and colleagues [72] reported functional differences between the left and the right FFA. Frässle and colleagues [73] showed that the size of the right OFA (but not the left OFA) was associated with the degree of asymmetrical cross-hemispheric recruitment of the face-selective areas. Furthermore, Proverbio and colleagues [74] showed the occurrence of predominantly right-sided event-related potentials (ERPs) related to facial emotion recognition in men, whereas the ERPs where detected bilaterally in women. This suggests an interaction between lateralization of face-related cognition and gender. In the light of these studies, we estimated all the models encompassing CT measures described below twice, once with right-sided and once with left-sided ROIs. Considering the results of the previous studies, we expected to find stronger relations in the right hemisphere. This would be reflected in higher intercepts and potentially stronger factor loadings on the face-specific nested factor of CT. In the case of higher factor reliability (in terms of stronger loadings), regression weights between face specific CT predicting behavioural performance were also expected to be higher in the right hemisphere. Although we expected these relations to be stronger in the right hemisphere, there was no indication in the hitherto literature to believe that they would be absent or reversed in the left hemisphere.

### 3.3.2. Dealing with potential confounds through sex, handedness and family relationships

The HCP data include left-handed participants of both sexes and participants in family relationships, such as parents and offspring or twins. To ensure that handedness does not impact the effects that will be reported in the study, all analyses were repeated by regressing all indicators on handedness. In order to identify any possible differences between males and females, we then split the sample according to the participants' sex and repeated all the previous analysis steps in a multiple group analysis with sex as grouping variable. Because investigating sex differences belongs to a side question, we did not postulate any specific hypotheses with respect to sex differences, but there was reason to believe that males and females differ in their general cortical structures [75] and with respect to their face cognition abilities [76,77],[3] making the exploratory analysis a worthy endeavour. With respect to the CT and performance relationship, sex differences were not explored in previous research. Thus, we considered an exploratory analysis in our study reasonable.

Finally, to control for the nested data structure through family relations, an analysis approach was needed which was able to deal with the clustered data. Such an option is available in M*plus*: the ANALYSIS=COMPLEX command was used to provide correct standard errors without estimating within and between cluster coefficients. For this, the family dependencies were specified in the analysis by the addition of a clustering variable containing information on family relationships. This technique is able to account for standard error bias that may be induced through the dependency of

---

[3]In the accepted protocol, the references were incorrectly formatted.

observations. Thus, using this approach we computed adjusted standard errors and the adjusted chi-square test for model fit [78]. This method has been applied, evaluated and established in different studies [79,80]. Because the family relations are not subject to our hypotheses and because only part of the sample includes observations that are clustered by family relations, the approach of controlling nestedness appeared more appropriate than multi-level modelling.

### 3.3.3. ROI extraction

A new multi-modal cortex parcellation by Glasser and colleagues [81] was taken as a basis of ROI analyses. The parcellation makes use of various brain measurements administered in the HCP sample, including myelin, cortical thickness, functional connectivity and task fMRI contrasts. It separates the brain into a total of 180 parcels. These small parcels are grouped into 22 larger sections according to their anatomical and functional similarities. When calculating CT measures for the general factor of CT, the larger sections (consisting of multiple parcels) were used for the calculation of mean CT values within each of the sections.

#### 3.3.3.1. ROIs to indicate the general factor of CT

The general factor of CT was thought to account for variance in CT shared across the cortex within a person as a general neurobiological feature of an individual brain. Therefore, the ROIs indicating this factor were chosen in a manner that they cover larger sections of the cortex which are functionally not related. This contrasted with the smaller, face-specific ROIs, used in a later step to explain additional variance shared between face-selective brain areas. To this end, all the parcels encompassed in a selection of the larger sections identified by Glasser and colleagues [81] were taken into account. The average CT values in each of these larger sections were used as indicators.

To choose the ROIs used to parametrize general cortical thickness, we followed two main criteria. The first criterion was the reliability of cortical thickness measures in the respective cortex area. We quantified reliability by two approaches. The first approach was the feasibility of manual tracing: assuming that manual tracing is only feasible in areas whose anatomical properties allow it, and that the same anatomical properties facilitate algorithmic CT calculation, we favoured regions in which manual tracing is still done [82]. Furthermore, Liem and colleagues [83] provide cortex maps of intra-class correlation coefficients (ICC) as an indicator of reliability for different FreeSurfer measures, including CT. The reliability criterion was reached if a region conformed to one or both of these approaches. To ensure the judgement of high reliability of CT measures, we made use of HCP pre- and post-test data of structural measures that are available for a sub-sample of 46 twins. The reliability estimations on the basis of manual tracing feasibility and the ICC according to Liem and colleagues [83] was confirmed by using these data for determining test–retest reliability of cortical thickness in the respective ROIs.

The second criterion for the choice of appropriate regions to include in the general CT factor pertained to the relationship of the respective areas with face cognition. CT in the core regions of face processing [20] indicated a face-specific CT factor. Above these three core regions, however, other brain areas have been reported as showing face-selective functional activity. Duchaine & Yovel [71] propose an alternative framework of face cognition, including additional face-selective regions in the inferior frontal cortex and the temporal cortex. We strived to exclude the respective areas from our analyses to avoid masked and uncontrolled face-specific variations in cortical thickness. Finally, we included five of the 22 larger regions of the cortex in the factor of general CT (see electronic supplementary material, supplement 2): the primary and secondary visual area (which we will further treat as one ROI, named Vis), the premotor cortex (PM), the superior parietal cortex (SPC) and the dorsolateral prefrontal cortex (DLPFC). All the major lobes were represented in the chosen, distributed sections with the exception of the temporal lobe. With the high involvement of the temporal lobe in the processing of visual stimuli, the danger of recruiting face-selective structures for indicating the general CT factor was deemed too high. Notably, this selection of ROIs was theory-driven and remained unchanged in later steps of the analysis.

#### 3.3.3.2. ROIs to indicate the face-specific CT factor

The locations of brain regions show a considerable between-subjects variability, particularly in areas relevant for higher cognitive functions [84,85]. Furthermore, McGugin and colleagues [39] found a relationship between CT in the FFA and performance in face-related tasks after localizing each

subject's individual FFA using functional MRI scans. Thus, we aimed to determine the individual location of FFA, OFA and pSTS by evaluating functional activity across the cortex. The localization was accomplished in three steps. First, we used the parcellation by Glasser and colleagues [81] to select *a priori* cortex areas as an across-subjects ROI (all parcels that served as ROIs are listed in electronic supplementary material, supplement 2). The selection of the appropriate parcels was done on the basis of previous functional MRI studies [86], reporting coordinates of peaks in face-selective activity in MNI space that were mapped to the respective parcels according to Glasser and colleagues [81]. Within this across-subjects ROI, the maximum activity for each subject in the respective contrast was detected.

In a second step, the maximum in blood-oxygen-level dependent (BOLD) signal during a face-related task within this across-subjects ROI was detected for each subject individually. In their distributed network approach of face processing, Haxby & Gobbini [20] suggest a functional distinction between FFA and OFA on the one hand, and pSTS on the other hand. FFA and OFA are thought to encompass processing of invariant facial features, thus being more involved in judging facial identity, whereas the pSTS is said to be rather associated with the variant facial features, such as facial expressions. Therefore, the contrast for localizing FFA and OFA was different from the contrast used for localizing the pSTS. FFA and OFA were localized by investigating the activity in the faces–average contrast of the working memory tasks (where average refers to the signal averaged across all other object categories used for the task, which were body parts, places and tools). The pSTS was localized by investigating the faces–shapes contrast in the emotion processing localizer tasks, where emotional faces and abstract shapes where used as stimuli. To ensure that the maximum activity was also substantial, a threshold of $t > 3$ was applied [87]. In preliminary tests on a sub-sample of $N = 10$ subjects, the contrasts were suited to detect above-threshold activity in the expected regions, indicating that the contrasts were able to localize the respective ROIs.

Finally, a circular individual ROI was drawn concentrically around the maximum. The radius was set to 5 mm. Average cortical thickness was extracted from these individual ROIs. Additionally, we overlaid the individual masks of all subjects and provide the locations of the ROIs that have proven to be the likeliest within the present sample. The across-subjects ROI locations are summarized in electronic supplementary material, supplement 3.

### 3.3.4. Test–retest reliability of CT measures

To ensure that the CT measures in the brain regions used to parametrize the latent factors of CT were reliable, we made use of the small number of subjects from the HCP sample for which retest data of 3T MRI modalities are available. This sub-sample consisted of 46 monozygotic twins. ICC coefficients of the CT estimates in the brain areas that we used in our analyses were obtained between the first and the second measurement.

## 3.4. Structural equation modelling

SEM is a statistical analysis framework modelling the pattern of covariances between multiple observed variables [88]. The model fit is generally evaluated by a number of different fit-indices with available conventions regarding their magnitude [89]: the comparative fit index (CFI > 0.95), the root mean square error of approximation (RMSEA < 0.08) and the standardized root mean square residual (SMMR < 0.05). Missing values in the final data were treated by using the full information maximum-likelihood method (FIML) built into M*Plus*. When observing clustered variables in M*Plus*, robust maximum-likelihood estimation (MLR) [90] has to be used instead of the usual maximum-likelihood (ML) estimation.[4] MLR estimation is based on robust standard errors [91], yielding an estimation procedure that is not reliant on the assumption of normal distribution. Therefore, testing for univariate normal distribution of the dependent variables was obsolete in this case. MLR estimators are a common tool for dealing with skewed data in SEM.

In the SEM analyses for hypothesis testing, nested models were compared. The structural models including brain and behavioural measurement models were tested successively. First, we tested a model only consisting of a general factor of performance accuracy and of a general factor of cortical thickness. In a second model, a face-specific factor was nested under the general factor for CT and performance, thus at both sides of the structural model. If the addition of the face-specific factors

---

[4]See §3.5, *Deviations from protocol*.

**Table 2.** Measurement and structural model of performance accuracy and CT. $\Delta\chi^2$ ($\Delta$d.f.): nested models were compared with the corresponding general factor model by using the $\chi^2$-difference test after Satorra–Bentler scaling correction [92]. accG—general factor of performance accuracy; accF—nested factor of performance accuracy in face-related tasks; CTG—general factor of cortical thickness; CTF—nested factor of CT in face-related brain areas; acc + CT—these models represent a combination of the respective accuracy and CT models (figure 3); R—indicates that right-hemispheric ROIs were used in the model; L—indicates the use of left-hemispheric ROIs.

| model | $\chi^2$ | d.f. | CFI | RMSEA | SRMR | $\Delta\chi^{2\ a}$ ($\Delta$d.f.) |
|---|---|---|---|---|---|---|
| accG | 451.01 | 76 | 0.80 | 0.08 | 0.06 | — |
| accG + accF | 307.34 | 67 | 0.88 | 0.07 | 0.05 | 110.66[b] (9) |
| CTG L | 173.83 | 26 | 0.93 | 0.08 | 0.05 | — |
| CTG + CTF L | — | — | — | — | — | — |
| CTG R | 318.97 | 26 | 0.87 | 0.12 | 0.06 | — |
| CTG + CTF R | 110.54 | 21 | 0.94 | 0.07 | 0.03 | 176.97[b] (5) |
| acc + CT L | — | — | — | — | — | — |
| acc + CT R | — | — | — | — | — | — |

[a]Satorra–Bentler corrected $\chi^2$-difference reported.
[b]Significant at an alpha level below 0.01.

increased model fit significantly, the nested factor was considered necessary and face specificity was concluded. Additionally, in order to test explicitly whether a specific nested factor was identified, we tested whether its variance was statistically different from zero. In order to do so, the latent variables were scaled by one of their indicators (see model descriptions below). The fit of all models and inferential comparisons between the models based on their likelihood are summarized in table 2. Finally, only if the two nested-factor models turned out to be superior to the models containing only specific factors in terms of fit, we planned to combine the two models into one structural model to test whether the individual differences in general and face-related performance could be predicted by individual differences in CT. Because trial-level behavioural performance is not available for all HCP tasks, McDonald's Omega [62] as a reliability coefficient cannot be provided for every task.[5] Thus, we reported Omega as a measure of reliability for every latent variable, which quantifies true score variance.

### 3.4.1. Measurement models of the behavioural data

Importantly, the measurement models of behavioural performance data were completely analogous to the models described above in §3.2 *Preliminary analyses of the behavioural performance data* (see figure 1 for model illustration). In a first model named acc1 (see figure 1*a*), a general factor of cognitive performance accG was established, parametrized by the accuracy rates recorded in all tasks. Next, we tested a second, nested model (see figure 1*b*). This second model included above the general factor of accuracy, accG, a second factor of face-specific performance accuracy, accF, nested under accG and orthogonal to it. This nested factor was indicated by accuracy in facial tasks only. In the case of the general factor of performance accuracy, the reference indicator was the performance in the outside-scanner working memory task. For the nested factor of face cognition accuracy, we used the inside-scanner working memory task as reference. In the case of the behavioural models, some of the indicators represented repeated measurements of the same tasks. This concerned the inside-scanner working memory task (both 0-back and 2-back conditions) and the recognition memory task, which were both performed during or after two separate scanner sessions. We tested whether we needed to account for shared variance in the repeated measures by adding a third model to the model sequence where the factor loadings of the two indicators representing the same tasks on two occasions were constrained to equality. If the model fit was not significantly changed by this addition, the more

[5]See §3.5, *Deviations from protocol*.

parsimonious model with[6] the additional constraints was carried over to the structural model linking behavioural performance and structural measures. We report model fit and model comparisons in table 2. Factor loadings are reported in electronic supplementary material, supplement 5.

### 3.4.2. Measurement models of CT

Before including all CT values into the SEMs, we residualized them for intracranial volume within each person. Furthermore, age and gender of each participant entered the measurement models as covariates that were controlled for. The models described in the following were tested twice, first using right-hemispheric ROIs and then by means of left-hemispheric ROIs.

In analogy to the measurement models of task performance, we tested two measurement models of cortical thickness. In the first model (see figure 2a), a general factor of CT, thus CTG, was estimated. CTG was indicated by the measured CT controlled for intracranial volume in a number of distributed brain areas, as indicated above (see the paragraph *ROI extraction* in the *Analysis and results* section). In a second step (see figure 2b), the model was extended by encompassing a further nested factor of CT in face-related areas, CTF besides the general factor CTG. CTF was parametrized solely by the CT measured in the three core regions of face processing. For the general CT model, the reference indicator was the CT measures within the primary and secondary visual cortex. In the case of the nested model of CT in face-related areas, CT in the FFA represented the reference. If there was specific systematic variance in CT shared between the three face-selective areas (hypothesis 1) of the core face network, then this nested factor was expected to account for a substantial proportion of variance above the general factor of CT. Thus, all three factor loadings of the nested face factor were expected to be significant in the second measurement model including CTG and CTF. Furthermore, the model fit was expected to increase substantially with the inclusion of the nested face factor. Model fit and comparison of the nested models are summarized in table 2. Factor loadings are reported in electronic supplementary material, supplement 6.

### 3.4.3. Structural model on the brain and behaviour relationship

In order to bring behavioural and neuroanatomical measures together, we planned to estimate a structural model combining the two nested models established in previous steps. This allowed us to test the hypothesized latent relationships between the established brain and behaviour factors (hypotheses 2 and 3, figure 3). Because there were two nested models of CT, encompassing measures either in the right or the left brain hemisphere, the structural model was also planned to be estimated for the right-hemispheric and the left-hemispheric CT measures.

In this structural model we planned to first investigate whether the latent factors of performance accuracy can be predicted by the latent factors of CT. General accuracy in cognitive tasks (captured by the latent factor accG) was expected to be predicted by the variance in CT across various regions of the cortex (captured by the general latent factor of CT, CTG; hypothesis 2; see figure 3). Thus, to address hypothesis 2, the regression path originating from CTG and directed on accG was the parameter of interest. Second, the accuracy in performance tasks using faces (captured by the latent factor accF) was expected to be predictable by the variance in CT measures of the core regions of face cognition (captured by the latent factor CTF; hypothesis 3). Therefore, the regression path originating from CTF and directed onto accG was to be inferentially tested against zero. Furthermore, we expected the regression coefficients within corresponding content domains (i.e. accG on CTG and accF on CTF) to be higher than the relationships between each content domain and its respective counterpart (i.e. accG on FCT or accF on GCT,[7] hypothesis 4). The postulated structural model and the hypothesized latent-factor relationships (the strength of each respective relationship indicated by the thickness of the line) are depicted in figure 3. The model fits of right- and left-hemispheric versions of the structural model are summarized in table 2. Factor loadings are reported in electronic supplementary material, supplement 7. Importantly, we planned to estimate this model only if, as a prerequisite, the face-specific nested factors of CT as well as performance turned out to be identified.

---

[6]In the accepted protocol, this sentence read: 'If the model fit was not significantly changed […], the more parsimonious model without the additional constraints was carried over […]' This was contradictory, because the more parsimonious model is the one with constraints. The inconsistency was corrected upon resubmission at stage 2.

[7]An error in the accepted protocol ('GCT on accF or FCT on accG') was corrected to 'accG on FCT or accF on GCT'. Behaviour is regressed on CT.

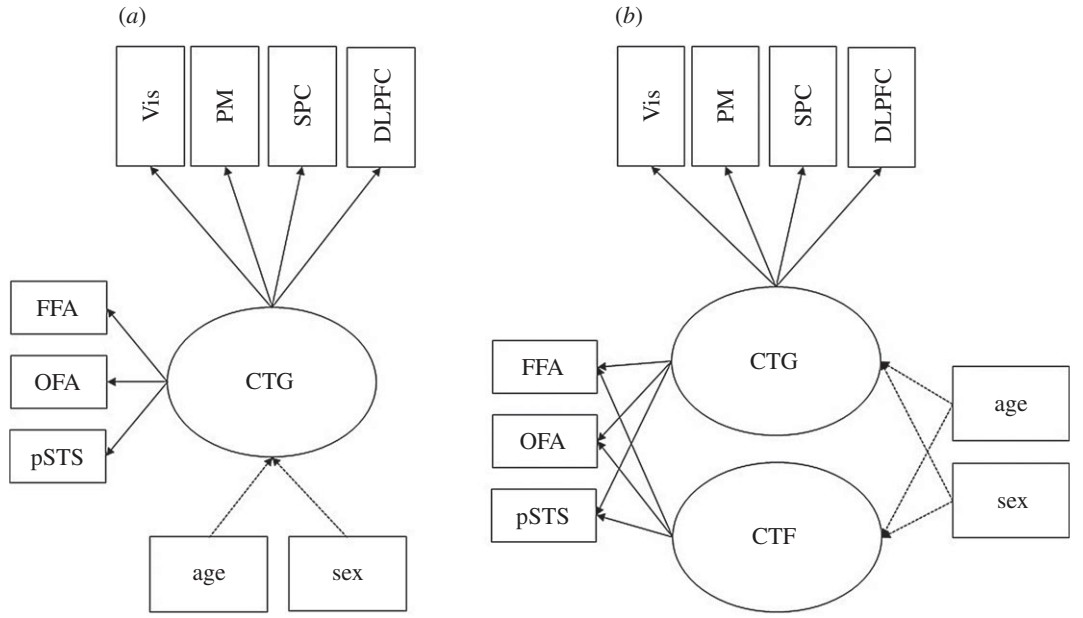

**Figure 2.** (*a,b*) Measurement models of CT in distributed and face-related brain areas. CTG—general factor of cortical thickness; CTF—nested factor of CT in face-related brain areas. Indicated by local cortical thickness in brain regions: Vis—primary and secondary visual cortices; PM—premotor cortex; SPC—superior parietal cortex; DLPFC—dorsolateral prefrontal cortex; FFA—fusiform face area; OFA—occipital face area; pSTS—posterior superior temporal sulcus. Age and sex are controlled for by establishing them as covariates (dotted lines indicate covariate influence on the latent factors that will be expected).

## 3.5. Deviations from protocol

From the preregistered protocol to the stage 2 submission, corrections were necessary. First, we originally planned to provide McDonald's Omega [62] as a measure of reliability for every indicator separately. However, at that stage, it was not clear yet whether we could acquire trial-level data for all tasks. It turned out that some tasks that we use as indicators for the general performance factors are only available from the HCP project as scores, not as trial data. Additionally, some tasks are based on adaptive testing strategies. Thus, in order to provide a measure of reliability that is consistent for the behavioural data, we decided to report the coefficient Omega as a measure of reliability for the latent factors. Second, upon submission of the stage 1 manuscript, we did not include age and sex as covariates in figure 1, which was corrected at stage 2. Furthermore, the behavioural performance indicators FRec 1 and FRec 2 (recognition memory of faces after scan 1 and scan 2) were missing in figure 1. The tasks were described in §2.3.1 *Outside-scanner psychometric tasks*, but forgotten in the figure. This change is also reflected in the figure captions. Third, we preregistered to use maximum likelihood (ML) to estimate the SEMs and use MLR estimation only if the indicators were non-normally distributed, which we planned to test using a Kolmogorov–Smirnov (KS) [93] test. However, MLR estimation has to be used in any case when clustered variables are observed in M*Plus*. Thus, ML estimation was not an option and the KS test was obsolete. Furthermore, when using MLR estimation, it is necessary to apply the Satorra–Bentler scaling correction [92] prior to model fit evaluation. This change is also reflected in the table captions. The stage 1 manuscript in its unchanged form can be found on the corresponding OSF site (https://osf.io/p7c8z/).

## 4. Results

### 4.1. Measurement models of the behavioural data

In the first model, a single factor accG was assumed to account for variation in performance accuracy across all tasks (see table 2 for an overview of fit indices and electronic supplementary material, supplement 5 for an overview of all standardized factor loadings and standard errors). Additionally, the latent factor was regressed onto age and sex. As expected, the fit of this model was not satisfactory: $\chi^2_{76} = 451.01$, CFI = 0.80, RMSEA = 0.08, SRMR = 0.06. Standardized factor loadings on

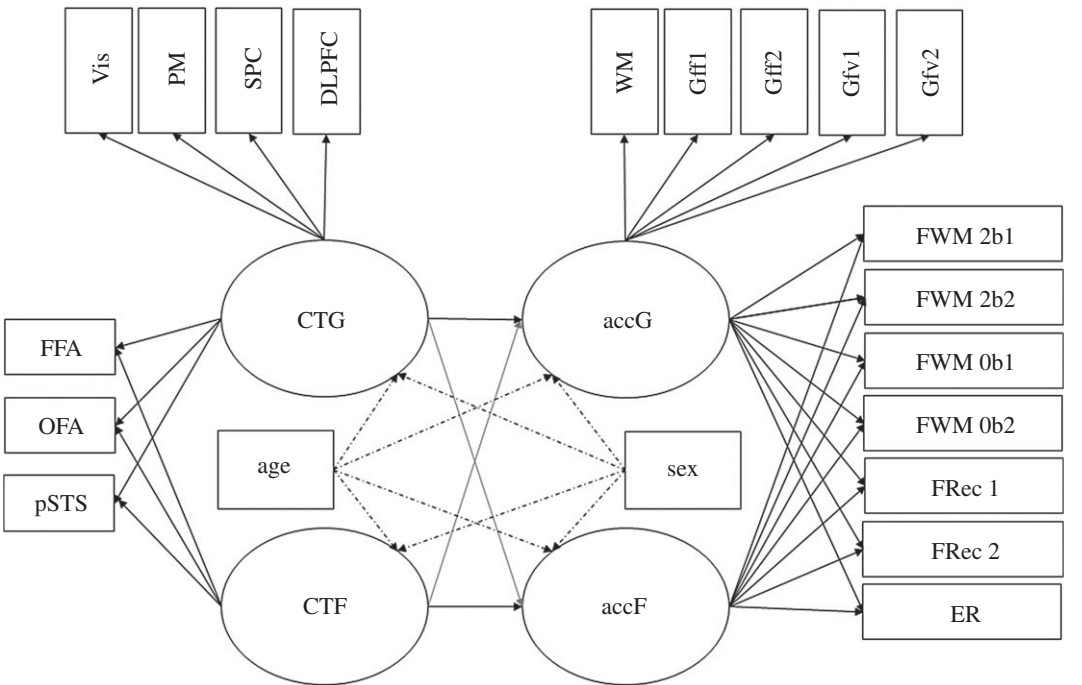

**Figure 3.** Postulated structural model for investigating differential relationship between CT and face perception accuracy as compared with general CT and general cognitive performance. Line thickness and colour intensity indicate the expected strength of the latent relationship, where black and thick lines indicate stronger postulated relationships than thinner, grey lines. accG and accF—latent variables accounting for performance accuracy in general and face-specific behavioural tasks. Indicators: WM—working memory; Gff1—figural task, progressive matrices; Gff2—figural task, spatial line orientation; Gfv1—verbal task, oral reading recognition; Gfv2—verbal task, vocabulary comprehension; FWM—working memory task with facial content in a 0-back and a 2-back condition; FRec—recognition memory of faces from the inside-scanner working memory task; ER—facial emotion recognition. CTG and CTF—global and face-specific cortical thickness. CTG is indicated by local cortical thickness in the following brain regions: Vis—primary and secondary visual cortices; PM—premotor cortex; SPC—superior parietal cortex; DLPFC—dorsolateral prefrontal cortex; FFA—fusiform face area; OFA—occipital face area; pSTS—posterior superior temporal sulcus. Age and sex are controlled for by including them as covariates (dotted lines indicate expected covariate influence on the latent factors).

accG ranged between 0.20 and 0.78 ($p$-values $\leq 0.001$). There was a statistically significant sex effect on accG ($\beta = 0.34$, $p < 0.001$; for an overview of standardized regression weights, see electronic supplementary material, supplement 4), whereas there was no age effect in this rather age-homogeneous sample ($\beta = -0.01$, $p = 0.44$). Adding the nested accF factor yielded a better model fit, although the model still did not describe the data very well: $\chi^2_{67} = 307.34$, CFI = 0.88, RMSEA = 0.07, SRMR = 0.05. Only accG was predicted by sex ($\beta = 0.38$, $p < 0.001$). Whereas the factor reliability of accG was rather high ($\omega = 0.76$), the nested factor accF showed a much lower reliability ($\omega = 0.32$), which was also visible in the low and non-significant loadings on accF (standardized loadings between 0.004 and 0.065, $p$-values up to 0.94; see electronic supplementary material, supplement 5). The residual variance of accF after controlling for the covariates was not significant ($\sigma^2_{\text{accF}} < 0.001$, $p = 0.369$), indicating that the factor was not identified. To assess potential improvement of model fit by adding the nested factor, the $\Delta\chi^2$ was adjusted using the Satorra–Bentler scaling correction [92]. The model with the face-related factor had a better fit and thus accounts for the observed data in a more appropriate (albeit not satisfactory) manner (corrected $\Delta\chi^2 = 110.67$, $p < 0.001$). When constraining the loadings of indicators captured by repeated measurements (see above) to equality, the fit did not diminish. Because the model fit was not satisfactory, we concluded that the final model structure as preregistered was not appropriate and data-driven explorations are justified to find suggestions for a potentially better solution (see §5.1).

## 4.2. Measurement models of CT

As preregistered, we set up the measurement models of CT in analogy to the accuracy models, starting with a single latent factor model (CTG), indicated by CT in all cortex areas. The model revealed

reasonable fit for the left hemisphere data: $\chi^2_{26} = 173.83$, CFI = 0.93, RMSEA = 0.08, SRMR = 0.05. Standardized loadings ranged between 0.25 and 0.91 ($p$-values $\leq 0.001$). CTG was associated with age in a statistically significant way ($\beta = -0.05$, $p < 0.001$), but not with sex. Interestingly, the fit was considerably worse in the right hemisphere: $\chi^2_{26} = 318.94$, CFI = 0.87, RMSEA = 0.12, SRMR = 0.06. Here, standardized loadings ranged between 0.51 and 0.84 ($p$-values $\leq 0.001$). In the right hemisphere, age ($\beta = -0.05$, $p < 0.001$) and sex ($\beta = -0.17$, $p = 0.05$) were related to CTG in a statistically significant way. When adding a nested factor CTF indicated by CT in the face-related ROIs, the left-hemisphere model did not converge. In the right hemisphere, however, the model fit improved: $\chi^2_{21} = 110.54$, CFI = 0.94, RMSEA = 0.07, SRMR = 0.03. The residual variance of CTF in the right hemisphere after controlling for the covariates was not significant, indicating that the factor was not identified ($\sigma^2_{\text{CTF}} = 0.001$, $p = 0.56$). Again, age was significantly related to CTG ($\beta = -0.06$, $p < 0.001$).

The loadings on the nested face-specific factor were not significant in the right hemisphere model (standardized loadings between 0.07 and 0.22; $p$s > 0.1; see electronic supplementary material, supplement 6 for all standardized factor loadings and standard errors). This was not surprising, given that the manifest correlations between CT measures in the face-related areas (i.e. the correlations between face-related ROIs within the same hemisphere or across hemispheres) were strongly heterogeneous (ranging between $r = 0.01$ and $r = 0.31$). In comparison, correlations of CT values in the regions indicating the general factor of CT were moderate to high, reaching up to $r = 0.80$ (between DLPFC and PMC in the left hemisphere). As reflected in the low factor loadings, the nested factor reliability of CTF was strongly restricted ($\omega = 0.06$) compared to the general factor CTG ($\omega = 0.75$).

We further explored this inconsistency of associations between general and face specific CT by conducting reliability analyses of the CT measures (see §4.4). Furthermore, we decided to further explore specificity of individual differences in CT of the face-related areas by increasing the size of the ROIs around the functional peak of individual subjects (see §5.2).

## 4.3. Structural models of the brain and behaviour relationship

The structural models as they were preregistered could not be estimated, since the fit of the measurement models of the behavioural as well as the CT data was not satisfactory and the models had to be adapted in exploratory analyses. We thus investigated brain–behaviour relationships after performing *post hoc* adaptations at the level of the measurement models (see §5.3).

## 4.4. Reliability of CT measures

We calculated one-way fixed effects ICCs based on the retest MRI data of 45 twins who were scanned repeatedly on two separate measurement occasions in the HCP project. This was accomplished with the psych package for R [94], which applies the ICC according to the approach by Shrout & Fleiss [95]. The values were between 0.72 for the functionally localized right pSTS and 0.85 in the right OFA. In comparison, the ICCs for the cortex wide areas were between 0.94 for the right PM and 0.97 for the right dlPFC and the right SPC (see table 3 for an overview of all ICCs). This discrepancy is further addressed in §5.4.

# 5. *Post hoc* analyses

Because the measurement models of accuracy and CT as preregistered were not suited to yield sound and interpretable results, we performed *post hoc* modifications of the preregistered protocol. Results are reported as follows: the modified accuracy and CT models are described according to the same sequence as the preregistered models. Thus, the logic of extending one-factorial measurement models of accuracy or CT by adding a nested, face-related factor, remains the same. In the case of the accuracy models, however, indicators needed to be dropped in order to find a proper model solution. In CT, the ROI size was successively increased because the 5 mm radius ROIs yielded results that were not reliable enough and consequently not interpretable. Finally, we report the modified structural models testing brain–behaviour relationship accounted for potential confounding variables (sex, handedness).

**Table 3.** ICCs as reliability estimates of CT measures with confidence intervals ($\alpha = 0.05$, two-tailed).

| | ROI size 5 mm | | | ROI size 10 mm | | | ROI mask across-subject | | |
|---|---|---|---|---|---|---|---|---|---|
| | ICC | LB | UB | ICC | LB | UB | ICC | LB | UB |
| Vis left | | | | | | | 0.95 | 0.88 | 0.97 |
| Vis right | | | | | | | 0.95 | 0.90 | 0.97 |
| PM left | | | | | | | 0.95 | 0.90 | 0.97 |
| PM right | | | | | | | 0.94 | 0.88 | 0.97 |
| SPC left | | | | | | | 0.97 | 0.95 | 0.99 |
| SPC right | | | | | | | 0.95 | 0.91 | 0.97 |
| dlPFC left | | | | | | | 0.97 | 0.95 | 0.99 |
| dlPFC right | | | | | | | 0.95 | 0.89 | 0.98 |
| FFA left | 0.77 | 0.56 | 0.88 | 0.78 | 0.58 | 0.89 | 0.97 | 0.94 | 0.98 |
| FFA right | 0.82 | 0.66 | 0.83 | 0.80 | 0.62 | 0.81 | 0.97 | 0.93 | 0.97 |
| OFA left | 0.80 | 0.61 | 0.90 | 0.85 | 0.71 | 0.92 | 0.97 | 0.95 | 0.99 |
| OFA right | 0.85 | 0.72 | 0.92 | 0.89 | 0.78 | 0.94 | 0.97 | 0.93 | 0.98 |
| pSTS left | 0.74 | 0.50 | 0.86 | 0.75 | 0.53 | 0.87 | 0.97 | 0.94 | 0.98 |
| pSTS right | 0.72 | 0.47 | 0.86 | 0.83 | 0.67 | 0.91 | 0.96 | 0.93 | 0.98 |

## 5.1. Modified measurement models of the behavioural data

The poor model fit and the loading pattern indicated that a substantive proportion of variance in cognitive performance was not accounted for. After inspection of the loading pattern, we modified the models as follows: (i) we omitted the 0-back working memory task and the emotion recognition task. Both tasks were too easy and performance was at ceiling in many cases. Thus, the variance was strongly restricted. Also, because the tasks were extremely easy, we reasoned that the processes necessary for task performance were simple perceptual, automated processes which are supposedly non-specific for content (see *Discussion*). Furthermore, (ii) we allowed a residual covariance between the 2-back working memory task performance measured in session one versus two. Finally, (iii) because a strong reference indicator could not be identified, factor variances were fixed to one for scaling purpose and all factor loadings were freely estimated. Standardized factor loadings as estimated in this modified model sequence are provided in electronic supplementary material, supplement 5.

The one-factorial modified accuracy model did not have a good fit: $\chi^2_{42} = 213.68$, CFI = 0.89, RMSEA = 0.07, SRMR = 0.05. Standardized loadings ranged between 0.18 and 0.80 (*p*-values ≤ 0.001). The general accuracy factor accG was associated with sex ($\beta = 0.38$, $p < 0.001$), but not with age.

Adding the nested factor accF led to significant improvement (table 4) and acceptable fit: $\chi^2_{38} = 149.72$, CFI = 0.93, RMSEA = 0.06, SRMR = 0.04. Standardized loadings ranged between 0.20 and 0.80 for accG (*p*-values ≤ 0.001) and 0.10 and 0.50 for accF (*p*-values ≤ 0.03). Again, the factor reliability of accG was rather high ($\omega = 0.76$), whereas the nested factor accF showed a much lower reliability ($\omega = 0.24$). In terms of factor loadings, however, the nested factor was identified, because all loadings were statistically different from zero. Sex was associated with accG ($\beta = 0.40$, $p < 0.001$) and accF ($\beta = -0.44$, $p < 0.001$).

## 5.2. Modified measurement models of CT

As described above (see §4.2), correlations among CT measures of the face-related ROIs were low as compared with correlations among CT values across other cortex areas. This observation was similarly true for correlations between the CTs of the within hemisphere face-related ROIs, between each face-related ROI and their contralateral counterparts, and between the face-related ROIs and other, general ROIs. This heterogeneous correlation pattern was unexpected, given that CT measures from different

**Table 4.** Modified measurement and structural models of performance and CT. $\Delta\chi^2$ ($\Delta$d.f.): nested models were compared with the corresponding general factor model by using the $\chi^2$-difference test after Satorra–Bentler scaling correction [92]. accG —general factor of performance accuracy; accF—nested factor of face-related performance; CTG—general factor of CT (CT); CTF—nested factor of CT in face-related brain areas; acc + CT—these models represent a combination of the respective accuracy and CT models (figure 3); R—indicates that right-hemispheric ROIs were modelled; L—indicates the use of left-hemispheric ROIs; m—indicates that the model series was altered compared to the preregistered protocol and is part of the *post hoc* analysis.

| | model | $\chi^2$ | d.f. | CFI | RMSEA | SRMR | $\Delta\chi^2$ [a] ($\Delta$d.f.) |
|---|---|---|---|---|---|---|---|
| acc modified | accGm | 213.68 | 42 | 0.89 | 0.07 | 0.05 | 61.35[b] (4) |
| | accGm + accFm | 149.72 | 38 | 0.93 | 0.06 | 0.04 | |
| ROI 10 mm (see §5.2.1) | CTGm L | 123.71 | 25 | 0.95 | 0.07 | 0.04 | 17.08[b] (5) |
| | CTGm + CTF L | 107.39 | 20 | 0.96 | 0.07 | 0.04 | |
| | CTGm R | 95.65 | 25 | 0.96 | 0.06 | 0.04 | 11.89[b] (5) |
| | CTGm + CTFm R | 84.14 | 20 | 0.96 | 0.06 | 0.03 | |
| ROI mask across-subject (see §5.2.2) | CTGm L | 187.67 | 25 | 0.95 | 0.09 | 0.05 | 31.41[b] (5) |
| | CTGm + CTFm L | 157.12 | 20 | 0.96 | 0.09 | 0.04 | |
| | CTGm R | 199.31 | 25 | 0.92 | 0.09 | 0.05 | 38.78[b] (5) |
| | CTGm + CTFm R | 160.56 | 20 | 0.94 | 0.09 | 0.05 | |
| | accm + CTm L | 384.16 | 117 | 0.94 | 0.05 | 0.04 | |
| | accm + CTm R | 402.91 | 117 | 0.93 | 0.05 | 0.04 | |

[a]Satorra–Bentler corrected $\chi^2$-difference reported.
[b]Significant at an $\alpha$ level below 0.01.

areas are usually highly related within individuals. This finding raised doubts concerning the reliability of the functional localization.

We therefore performed an exploratory analysis by successively increasing the ROI size (see figure 4 for visualization of the three mask sizes). Initially, as preregistered, ROIs consisted of spherical structures with a 5 mm radius around the vertices where the maximum BOLD activity during a localizer task was recorded. In *post hoc* analyses, we increased the ROI radius to 10 mm. Additionally, we measured CT on the basis of the larger, non-individual parcels taken from the parcellation by Glasser *et al.* [81]. Using the CT data acquired with these alternative ROI sizes, the model sequences described previously (see §4.2) were re-estimated. Besides increasing the ROI size, we applied two further modifications: (i) we allowed for a residual covariance between the dlPFC and the PM (considering the close proximity of these two regions) and (ii) the factors were standardized for identification purpose, allowing all factor loadings to be freely estimated.

### 5.2.1. Modified measurement models of CT with individual ROIs of 10 mm radius

Using CT from ROIs of 10 mm radius, we estimated the above-described series of measurement models (see electronic supplementary material, supplement 6 for all standardized factor loadings). The first measurement model estimated one general factor, CTG. The fit of this model was quite good in the case of the left hemisphere: $\chi^2_{25} = 123.705$, CFI = 0.95, RMSEA = 0.07, SRMR = 0.04. Standardized loadings ranged between 0.36 and 0.85 (*p*-values $\leq$ 0.001). For the right hemisphere, the fit was even better: $\chi^2_{25} = 95.65$, CFI = 0.96, RMSEA = 0.06, SRMR = 0.04. Here, standardized loadings ranged between 0.37 and 0.85 (*p*-values $\leq$ 0.001). For both left and right hemisphere 10 mm ROIs, age ($\beta_{left} = -0.035$, $p \leq 0.01$; $\beta_{right} = -0.037$; $p < 0.01$) and sex ($\beta_{left} = -0.203$, $p = 0.02$; $\beta_{right} = -0.211$; $p = 0.02$) were significantly related with CT.

Next, we added a nested factor CTF to the models. The fit in the left-hemisphere model improved ($\chi^2_{20} = 107.45$, CFI = 0.96, RMSEA = 0.07, SRMR = 0.04). However, the nested face-related factor was not identified as suggested by the factor loadings, which were all non-significant (standardized estimates ranging between 0.02 and 0.50; *p*-values $\geq$ 0.21). The factor reliability of CTF was very low ($\omega = 0.18$) in contrast to a satisfactory reliability of CTG ($\omega = 0.80$) in the left hemisphere. The same pattern was

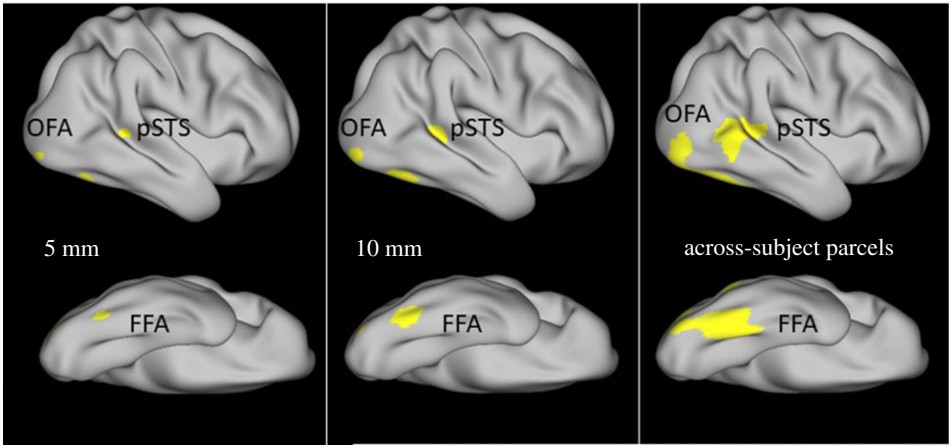

**Figure 4.** ROIs of three different sizes in a randomly selected subject. The figure illustrates the masks used to calculate CT values in the respective areas in the right hemisphere. Upper row: lateral view; lower row: ventral view. The upper and lower image on the left side depict ROIs with 5 mm radius; the images in the middle depict ROIs with 10 mm radius; the images on the right depict across-subject parcels used as masks.

found for the right hemisphere: adding CTF yielded an improved model fit ($\chi^2_{20} = 84.135$, CFI = 0.96, RMSEA = 0.06, SRMR = 0.03), but the factor was not identified (non-significant loadings with standardized estimates between 0.09 and 0.27 and $p$-values ≥ 0.17). The factor reliability of CTF was even lower in the right hemisphere ($\omega = 0.06$), while CTG showed a good reliability ($\omega = 0.78$). Again, based on these measurement models building upon CT measures in 10 mm diameter ROIs around the functional peak, no structural models of brain–behaviour relationships could be estimated. Manifest correlations between CT values were more homogeneous compared to the results from 5 mm sized ROIs. Still, however, the correlations between CT in face-related areas were lower (between $r = 0.14$ and 0.26) than between CT in general areas across the cortex (up to $r = 0.80$).

### 5.2.2. Modified measurement models of CT with across-subject ROIs

In addition to increasing the size of the individually localized ROIs, we calculated the mean CT for each subject within the parcels that were *a priori* deemed most likely to encompass the core network of face processing. These were the same parcels that served as masks for individual ROI localization (as described in §3.3.3, see electronic supplementary material, supplement 2). The correlations between CT in the face-related and in the general areas as well as among the CT values within the face-related areas were higher for across-subject ROIs (ranging between $r = 0.29$ and $r = 0.59$), but they still did not reach the magnitude of observed relationships between CT values in the general, more widespread cortex regions.

Using CT values from different across-subject ROIs as indicators, we again estimated a general model of CT in a first step (see electronic supplementary material, supplement 6 for an overview of all the standardized factor loadings). The model fit was acceptable in the left hemisphere: $\chi^2_{25} = 187.67$, CFI = 0.95, RMSEA = 0.09, SRMR = 0.05. Standardized loadings ranged between 0.63 and 0.79 ($p$-values ≤ 0.001). In the right hemisphere, a similar picture emerged: $\chi^2_{25} = 199.31$, CFI = 0.92, RMSEA = 0.09, SRMR = 0.05. Here, standardized loadings ranged between 0.56 and 0.78 ($p$-values ≤ 0.001). Age ($\beta_{\text{left}} = -0.031$, $p ≤ 0.01$; $\beta_{\text{right}} = -0.034$, $p < 0.01$) and sex ($\beta_{\text{left}} = -0.22$, $p = 0.01$; $\beta_{\text{right}} = -0.19$, $p = 0.04$) predicted the general factors CTG in the left and right hemispheres. When adding a nested factor CTF, in the left hemisphere, the fit improved: $\chi^2_{20} = 157.115$, CFI = 0.96, RMSEA = 0.08, SRMR = 0.04. Furthermore, the nested factor CTF was identified, as loadings of all face-related indicators were statistically substantial (standardized estimates ranging between 0.21 and 0.39; $p$-values < 0.001). The reliability of CTF was found to be low ($\omega = 0.26$), whereas the reliability of CTG was satisfactory ($\omega = 0.87$). In the right-hemisphere model, the fit improved as well by adding the face-specific factor: $\chi^2_{20} = 160.56$, CFI = 0.94, RMSEA = 0.09, SRMR = 0.05. Again, every loading on the nested factor was significant (standardized estimates ranging from 0.26 to 0.38; $p$-values < 0.001). Factor reliability was low for CTF ($\omega = 0.27$), but satisfactory for CTG ($\omega = 0.84$). Age ($\beta_{\text{left}} = -0.038$, $p_{\text{left}} < 0.01$; $\beta_{\text{right}} = -0.043$, $p = < 0.01$) and sex ($\beta_{\text{left}} = -0.23$, $p = 0.01$; $\beta_{\text{right}} = -0.22$, $p = 0.01$) were related with the general factors CTG left and right. Furthermore, the nested right-hemispheric factor CTF was related with age ($\beta_{\text{right}} = 0.04$, $p = 0.04$).

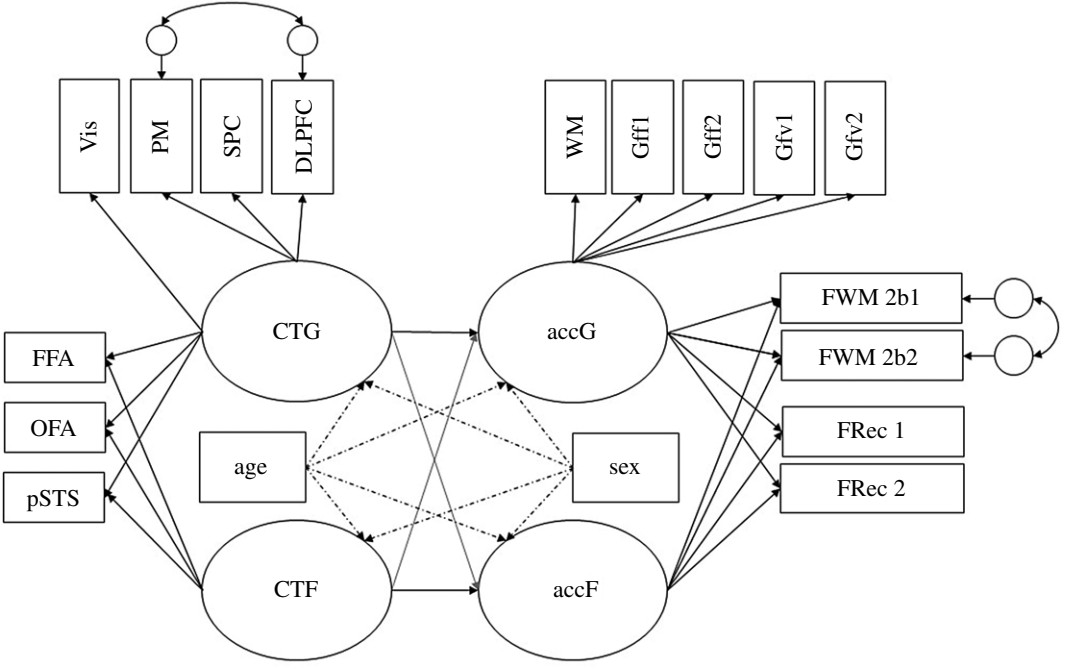

**Figure 5.** Modified structural model of brain–behaviour relationship. accG and accF—latent variables of accuracy in general and face-specific tasks. Indicators: WM—working memory; Gff1—figural task, progressive matrices; Gff2—figural task, spatial line orientation; Gfv1—verbal task, oral reading recognition; Gfv2—verbal task, vocabulary comprehension; FWM—working memory task in the 2-back condition; FRec—recognition memory of faces from the inside-scanner working memory task. CTG and CTF—global and face-specific CT (CT). CT indicators: Vis—primary and secondary visual cortices; PM—premotor cortex; SPC—superior parietal cortex; DLPFC—dorsolateral prefrontal cortex; FFA—fusiform face area; OFA—occipital face area; pSTS—posterior superior temporal sulcus. Age and sex are controlled for by including them as covariates (dotted lines indicate estimated regression paths).

## 5.3. Modified structural models of the brain and behaviour relationship

Using the CT data acquired from the across-subject ROIs (see above, §5.2.2) in both hemispheres, we combined the modified model of accuracy with the CT model (see figure 5 for visualization and electronic supplementary material, supplement 7 for all standardized factor loadings). The resulting structural model described the observed data acquired from the left hemisphere well: $\chi^2_{117} = 384.16$, CFI = 0.94, RMSEA = 0.05, SRMR = 0.04. For the accuracy factors in this model, standardized loadings ranged between 0.21 and 0.80 for accG ($p$-values $\leq 0.001$) and 0.10 and 0.50 for accF ($p$-values $\leq 0.03$). For the CT factors in this model, standardized loadings ranged between 0.58 and 0.83 for CTG ($p$-values $\leq 0.001$) and 0.25 and 0.41 for CTF ($p$-values $\leq 0.03$). For CT acquired from the right hemisphere, the fit was comparable: $\chi^2_{117} = 402.91$, CFI = 0.93, RMSEA = 0.05, SRMR = 0.04. In the right hemisphere, for the accuracy factors in this model, standardized loadings ranged between 0.19 and 0.80 for accG ($p$ values $\leq 0.001$) and 0.10 and 0.50 for accF ($p$-values $\leq 0.03$). For the CT factors in this model, standardized loadings ranged between 0.51 and 0.82 for CTG ($p$-values $\leq 0.001$) and 0.29 and 0.40 for CTF ($p$-values $\leq 0.001$). Among the regression weights of the latent performance factors onto the latent CT factors, only the relationship between general accuracy (accG) and the face-related factor of CT (CTF) was statistically significant in both hemispheres ($\beta_{\text{left}} = -0.26$, $p < 0.01$; $\beta_{\text{right}} = 0.23$, $p = 0.03$), indicating that accuracy across all tasks was positively associated with common individual differences of CT in the FFA, the OFA and the pSTS (see electronic supplementary material, supplement 4 for a comprehensive overview of all regressions). Surprisingly, CTG was not related with general accuracy.

## 5.4. Reliability of CT measures in the extended ROIs

Retest MRI data from 45 twins were available. We used these to calculate ICCs between the measured CT values at both measurement occasions (see also §4.4 and table 3). This procedure was repeated for the ROIs of 10 mm radius and the ROIs acquired from the across-subject ROIs in order to systematically

assess whether the size of the ROI affects the reliability of the CT measurement. Results indicate that CT measures acquired by the large across-subject ROIs based on the parcellation of Glasser *et al.* [81] produce more reliable results than functionally localized ROIs, although ICCs also increased when the ROI size was 10 mm instead of 5 mm.

As another, albeit more indirect, measure of reliability, we quantified the magnitude of relationship between the CT values acquired using the three competing CT measurement techniques. The most stringent way to observe the residual relationship between different CT measures would have been to correlate the latent factors indicated by 5 mm and 10 mm ROIs in a structural model. However, latent factors of CT in the face network were not identified in 5 mm or 10 mm ROIs. Therefore, correlations on manifest level are reported in electronic supplementary material, supplement 8. The correlation between CT values in the functionally localized areas are rather high, ranging between $r = 0.85$ and $r = 0.9$. In comparison, the relationship between CT values in the across-subject masks with localized ROIs range from $r = 0.36$ to $r = 0.57$.

## 5.5. Multiple group analysis for testing the brain and behaviour relationship across sex

To investigate whether brain–behaviour relations are moderated by sex, in the final modified models, we first tested for measurement invariance across the two groups of female and male participants. If SEM parameters are to be compared across groups, measurement invariance is a prerequisite. Only in the case of measurement invariance, the model parameters can be compared across groups, because then it can be assumed that latent variables have the same meaning in the case of both groups. Stepwise, we tested for configural, metric and strict measurement invariance while defining sex as a grouping variable [96,97]. The models tested for measurement invariance encompassed the modified accuracy model including a general and a face-related factor, the CT measurement models estimated with data from the large across-subject ROIs including the general and face-related factors of CT for both hemispheres, and the structural models of accuracy and CT for both hemispheres. All model fit parameters for the multiple group analyses are summarized and provided in electronic supplementary material, supplement 9. Because the $\chi^2$ difference test in multiple group analyses is not interpretable in MLR estimation, consecutive model constraints were statistically tested by means of the Wald test (see electronic supplementary material, supplement 9) [98]. In summary, for all models containing performance measures, metric measurement invariance could be established. In these cases, imposing equality constraints on the factor loadings across groups to test for metric invariance did not deteriorate the model fit significantly, but imposing equality constraints on the intercepts did. The models of CT revealed strict invariance, thus factor loadings and intercepts of the CT indicators had the same meaning across sex.

After establishing at least metric measurement invariance across all models, we constrained all factor loadings of the structural models across groups to equality and regarded the regression paths of the performance factors onto the CT factors in both sub-samples. The model described the data acquired from the left hemisphere sufficiently well: $\chi^2_{231} = 458.26$, CFI = 0.95, RMSEA = 0.05, SRMR = 0.05. Using the same model on data acquired from the right hemisphere, an acceptable model fit was revealed as well: $\chi^2_{231} = 513.96$, CFI = 0.93, RMSEA = 0.056, SRMR = 0.06. In both hemispheres, we found a positive significant relationship between the general factor of accuracy, accG, and the face-related factor of CT that was numerically different across sexes, depending on the hemisphere. In the left hemisphere, the regression of accG onto CTF was significant for males ($\beta = 0.32$, $p = 0.007$), but not for females ($\beta = 0.24$, $p = 0.15$). In the right hemisphere, the regression of accG onto CTF was significant for females ($\beta = 0.27$, $p = 0.03$), but not for males ($\beta = 0.13$, $p = 0.69$). This difference between males and females was further tested for statistical significance. To achieve this, we constrained the regression path from accG onto CTF to equality across sexes and tested whether this significantly deteriorated the model fit. This was not the case as revealed by the Wald test statistic (left: $p = 0.76$; right: $p = 70$), indicating that the small numerical difference between sexes was not significant. For details on fit indices, factor loadings and regression weights of the structural models conducted with male and female sub-samples, please see the outputs in our OSF project (https://osf.io/p7c8z/).

## 5.6. Influence of handedness on models of CT

In the right hemisphere, CT in the parcels encompassing the FFA and OFA were related with handedness. The effects were small ($\beta = 0.002$, $p = 0.007$) and positive, indicating that subjects with higher ratings in the handedness questionnaire (i.e. subjects who reported themselves as being more

reliant on their right hand) had a slightly thicker cortex in the right-sided FFA and OFA. After controlling for this effect in the final models of CT and performance, neither the fit or any of the loadings were significantly changed. For details on fit indices, factor loadings and regression weights of the handedness models, please see the outputs provided in our OSF project of this work (https://osf.io/p7c8z/).

## 5.7. Models combining both hemispheres

In order to bring both hemispheres together into one structural model, we followed two approaches. First, we collapsed the CT values for each ROI across hemispheres and computed a mean score of the left and right regions. As input data, we used CT values acquired with the large across-subject ROIs. With the mean CTs of each region, the structural model incorporating performance and CT factors was estimated again. The model fit was comparable with the previous models: $\chi^2_{117} = 440.14$, CFI = 0.93, RMSEA = 0.06, SRMR = 0.04. No new patterns of brain–behaviour relations were discovered with averaged CT across hemispheres. The general performance factor was significantly related with the latent factor of CT within the face-related areas ($\beta = 0.23$, $p < 0.01$), but no other effect was substantial.

Second, we included CT data from both hemispheres, again acquired using the large across-subject ROIs, into one model. Thus, not only did this structural model contain a general and a nested face-related factor of performance and CT, but there were two CT factors, one for the left and one for the right hemisphere. We allowed residual covariances between each of the CT measures with their respective counter-hemispheric equivalent. Furthermore, in addition to establishing orthogonality between the general and face-related factors of CT within each hemisphere, the paths from left general CT to the right face-related CT and vice versa were fixed to zero as well (the model syntax is provided in the OSF project of the present study, https://osf.io/p7c8z/). Finally, data-driven, further residual covariances were allowed in order to achieve acceptable model fit. These encompassed covariances of the both-sided PM with the both-sided dlPFC as well as the both-sided SPC with the both-sided visual cortex. These residual covariances are in line with the topographic proximity of the respective areas.

The model fit was satisfactory: $\chi^2_{233} = 440.14$, CFI = 0.95, RMSEA = 0.05, SRMR = 0.05. There were no significant relations between performance and CT. Notably, when all indicators were included in one model, the standard errors were much higher than in the single-hemisphere models.

## 5.8. Follow-up models separating indicators according to the content domains

In order to follow up on the unexpected finding that CT in face-related regions predicted the general factor of performance, but not the face-related factor of performance, we aimed for one further exploratory analysis, targeting the general and face-related indicators separately. A series of follow-up models was estimated, each containing a factor of performance accuracy predicted by a factor of CT (CTG predicting accG, CTG predicting accF, CTF predicting accG, CTF predicting accF; see figure 6). To investigate the factors in isolation, face-related indicators of performance and CT were omitted from the general factors for this specific analysis. In short, when estimated separately, none of the relationships between performance factors and CT factors were significant, including the regressions of accG on CTF factors. A table summarizing the fit indices and regression weights provided by these follow-up analyses is given in electronic supplementary material, supplement 10.

# 6. Discussion

We applied SEM to CT data to answer the question whether the factorial structure found consistently in behavioural data matches the structure of individual differences in CT. More specifically, we expected CT across widespread brain areas to covary and form a latent factor. In addition, we expected the three areas of the core face network to form an additional latent factor, nested under the general factor (hypothesis 1). Importantly, applying the accepted analysis protocol, no such specificity was detected. When functionally localized areas of FC were investigated, no specific latent factor was discovered, presumably due to reliability issues in functionally localizing rather small brain regions and the restricted specificity of CT in the face area, which can probably be discovered with highly reliable CT measures only. We found that *post hoc* adaptations to the preregistered analysis were necessary in order to ensure a higher reliability of CT measurement. After these unregistered adaptations were

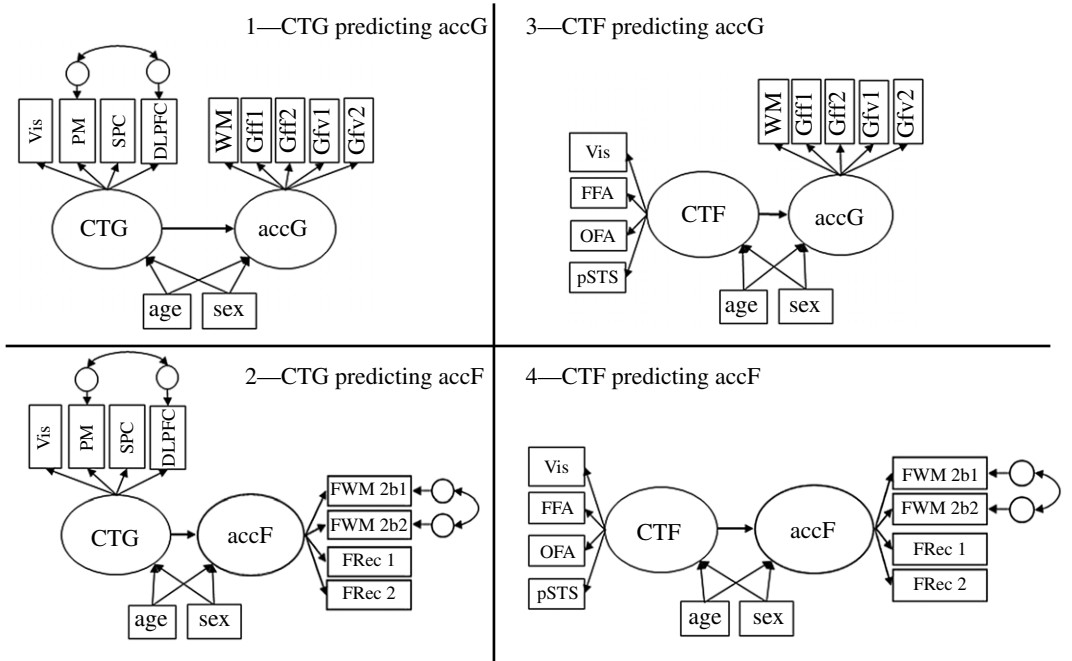

**Figure 6.** Follow-up model series separating general and face-related indicators. accG and accF—latent variables of accuracy in general and face-specific tasks. Indicators: WM—working memory; Gff1—figural task, progressive matrices; Gff2—figural task, spatial line orientation; Gfv1—verbal task, oral reading recognition; Gfv2—verbal task, vocabulary comprehension; FWM—working memory task in the 2-back condition; FRec—recognition memory of faces from the inside-scanner working memory task. CTG and CTF—global and face-specific CT (CT). CT indicators: Vis—primary and secondary visual cortices; PM—premotor cortex; SPC—superior parietal cortex; DLPFC—dorsolateral prefrontal cortex; FFA—fusiform face area; OFA—occipital face area; pSTS—posterior superior temporal sulcus. Age and sex are controlled for.

applied and CT measures acquired by using across-person parcels as masks were taken into account, the emerging latent factor structure resembled the factorial architecture observed in the behavioural data. It should be kept in mind, however, that most of the following interpretations are based on results acquired using a *post hoc* modified analysis. Therefore, they need to be confirmed with cross-validation. All adaptations to the preregistered protocol are discussed below (§6.1, *On the necessity of post hoc analyses*).

Furthermore, we investigated whether latent factors of CT discovered in exploratory *post hoc* analyses were associated with behavioural performance and, more specifically, whether the face-related factor of CT could predict performance in tasks with faces (hypothesis 2–4). Unexpectedly, our models demonstrated that within hemispheres, the factors of face-related CT predicted the general factor of accuracy or *g*. Thus, individual differences in performance were predicted by individual differences in CT, but the expected pattern of regressions did not emerge. Possible explanations are discussed below (§6.3, *On predicting stimulus specific cognitive performance by cortical thickness*).

## 6.1. On the necessity of *post hoc* analyses

We aimed to apply necessary changes to the preregistered protocol while adhering to it as closely as possible. Regardless of these efforts, the results reported here in this article are to a large part exploratory and should be interpreted as such. Although this restricts the interpretability of our findings until they are confirmed by cross-validation, we still think that the results acquired after *post hoc* modification of our analyses are important for future studies.

Firstly, in order to avoid problems with the performance data, we initially tested our models in a sub-sample of 200 participants prior to registration. However, despite this precautionary preliminary analysis, in the remaining main sample of 854 individuals, the model fit of the preregistered performance models was not satisfactory. Furthermore, the pattern of factor loadings was very heterogeneous, indicating that the models did not capture the data structure well. A substantive issue was that some of the tasks were not difficult enough. Specifically, in the multivariate distributions of the 0-back task and the Penn emotion recognition task [55], strong ceiling effects became apparent. Also, these task indicators did

not load onto the face-related performance factor, indicating that they did not share variance with the other face-related tasks above the general cognitive performance factor.

This pattern of findings can be explained in terms of dual-process theories of cognition [99]. These theories assume cognitive operations to vary qualitatively depending on whether they are simple and require rather automated processes as compared with higher, more complex cognitive performance that relies on deliberate processing. Simple and automated processes happen quickly, whereas more deliberate, top-down higher cognitive processes follow thereafter. Very simple tasks are solved relying on processes that are also termed elementary cognitive processes (ECPs) [100]. One possible explanation why face specificity in performance speed has been found in some newer studies [101–103], but not in others [6–9] pertains to this distinction between elementary and higher cognitive tasks. If a task is very simple, it is thought to tap into processes of perception, visual decoding and motor response. Evidence for this separation comes for example from studies analysing event-related potentials (ERPs), where early and automatic components were found not to be influenced by stimulus type for example [104,105], while later ERPs representing deliberate cognitive processing are subject to a large variety of influences [106,107]. The reason why the 0-back and ER tasks did not load onto the same factor as the other face-related tasks is possibly that they were too easy and tapped perceptive and automatic processes rather than higher face-related cognitive processes that cannot be conveyed by performance accuracy which was at ceiling.

Second, for the CT data, the correlation matrix of the measured variables seemed to be affected by measurement error. As a consequence of the unreliability of the localization and the rather low specificity of CT for dedicated brain networks, the measurement models of CT could not reflect the expected cortex structure. The low reliability of CT values acquired using small ROI sizes calls into question the reliability of the localizer task itself. The rationale underlying localizer tasks is that by contrasting BOLD activation recorded while a specific task or stimulus was given as compared with a contrast stimulus category, the results will reflect areas that are particularly involved in processing this specific stimulus category [108]. However, it is possible that localizer tasks from the HCP were not precise enough, as localization of particular regions has limitations [109]. For instance, Fox and colleagues [110] greatly improved the reliability of localizing the FFA, the OFA and the pSTS by using dynamic instead of stationary stimuli, which is a far better match to stimuli encountered in natural situations. Furthermore, the low correlations of the CT values acquired with the small ROIs could have emerged because CT computation is not equally good in every vertex. Calculating CT exactly at a position with strong cortex folding can impose a bias on the results acquired with the FreeSurfer algorithm [111,112]. Possibly, this issue was amplified when the ROIs were small. Comparably, when larger ROIs were used, more vertices were included in the average CT values so that measurement error could be better compensated. We found a divergence between the correlations among CT measurement using functionally localized ROIs of different sizes and the non-individual across-subject ROIs. The relationships between CT measurement using functionally localized ROIs and larger across-subject parcels were more divergent, showing correlations of medium strength at best.

## 6.2. On face specificity at the level of cortical thickness and behaviour

One central aim of the present work was to study the functional organization of the cortex. In previous studies on individual differences in structural and functional MRI or DTI data [40,113,114], latent variables in different neurobiological measures emerged. Likewise, we found a strong general factor of CT. Furthermore, going beyond those studies, we aimed to assess a domain-specific latent factor of CT as an integral and functionally meaningful morphological property of the face processing network. This aim was built upon the assumption that brain areas sharing a common functionality would often be employed simultaneously and thus in terms of neural plasticity be subject to common and specific morphological development as compared with functionally less related areas.

We could show in a large sample that, indeed, regions of the cortex with a common cause shared variance in terms of CT even after individual differences in CT across the entire cortex were accounted for. In other words, we provide exploratory evidence of a specific brain property that represents an individual's CT in the core face network. This suggests that CT is not only a property of a circumscribed area in a given brain, but a feature of a network of functionally related brain areas.

What is the specific functional interpretation of the latent factor of face-related cortical network thickness observed here? Variance in CT within and across individuals is determined by a large number of possible genetic, biological or pathological causes [115–117]. In addition, changes in grey matter as a consequence of neural plasticity have been discussed [34] as a mechanism of neural

learning. There are examples in the literature showing that CT is not exclusively altered in reaction to pathological or developmental changes, but also accompanying normal learning processes [47,118]. The finding of specific network-wide CT in the core network of FC provides evidence for a hitherto uninvestigated mechanism of neural plasticity in the brain that underlies functional differentiation of the cortex due to the social relevance of faces. It should be noted, however, that the proportion of variance specifically attributable to this network CT factor was small compared to the total variance in CT across all brain regions.

From a theoretical perspective, the interpretation of latent variable modelling approaches in neurocognitive psychology is portrayed in Kievit [119]. According to Kievit, likewise to the positive manifold on the behavioural level, morphological characteristics of the brain consistently correlate with each other. Thus, it is a fruitful endeavour to observe the interplay between dependent variables on the neural level. Accordingly, interrelations between morphological variables can be used to explain individual differences in behaviour. In other words, by defining and empirically consolidating properties at which brains may differ across people, we converge to a comprehensive understanding of the neural underpinnings of cognition. In the present study, we detected the outcome of a specific brain developmental mechanism that leads to functional differentiation of the cortex. The common use of brain areas for specific purposes goes along with common morphological development of these regions. In future studies, in order to investigate the relationship with respective abilities measured at the behavioural level, rigorous collection of performance data with tasks specifically tailored to capture individual difference in FC will be needed to overcome some of the limitations we faced.

## 6.3. On predicting stimulus specific cognitive performance by cortical thickness

Another major aim of the present study was to investigate whether individual differences in the ability to process faces were specifically related to CT in the core face network. Importantly, *post hoc* adaptations to the CT measurement and SEMs were necessary before any meaningful latent variables could be extracted, rendering the following results exploratory. Based on the modified measurement models, we estimated structural models to investigate whether we could predict general and face-related task performance by general CT across the entire brain or specific CT in the face network. A complex pattern of relationships emerged. When observing the hemispheres separately, individual differences in CT in the face network predicted general task performance, but not specific face-related performance. When data from the left and right hemispheres were averaged, the same pattern arose. In a next step, we dismantled the factors and observed the relationships between performance in general or face-related tasks onto the isolated factors of CT either in the general areas or in the face-related network. This way, we hoped to gain a better understanding of the causes of the regressions observed in the structural models. However, when isolated like this, general CT and face network CT predicted neither general nor face-related performance.

Two aspects of the present study should be viewed with caution when interpreting these exploratory results. The first is the reliability and validity of the face-related ability factor. Previous studies on the factor structure of FC [6,101,103] have often used extensive task batteries including easy and difficult tasks. FC is generally understood as a complex constituent ability of social cognition which is prerequisite to higher-order social processes [120–122]. Usually, the literature distinguishes between facets of FC, such as memory and perception of faces that have been shown to be rather independent components of FC [6,123], or the processing of variant versus invariant facial features [18,71,124], which corresponds to common views of the distinct functions in the core network of face processing. In the present study, ceiling effects and the very limited selection of face-related tasks compromise the interpretability of the face-related performance factor, which is also reflected in its very low factor reliability Omega [62] (see §§4.1 and 5.1). Also, with only very few tasks at hand that relied solely on small numbers of trials, we cannot distinguish between more fine-grained facets of FC. Note that in nested factor models, the reliability of the general latent factors is considerably higher than the reliability of the nested factors, because the reliability coefficient reflects the variance attributable to a specific factor. This necessarily makes it less likely to detect relationships between nested factors and other dependent variables. Because of these shortcomings, we refrain from interpreting the relationship between the face-related performance tasks and CT reported here too strictly. With the data at hand, hypotheses 2–4 were challenged. Concerning hypothesis 3 (CT in face network predicts performance in face tasks) and 4 (relationship between CT and performance is stronger within domains than across domains), more detailed research is needed in the future.

The second question that arises is how appropriately the general factor of CT reflects general variation across the cortex in areas related, to some extent, to cognition. In fact, the areas we chose to indicate the general CT factor were deliberately chosen due to criteria of measurement reliability [83]. However, the surprising finding that there is no relationship between general CT and cognitive performance in a number of general cognitive tasks suggests that more heed should be paid in future studies to selecting a functionally diverse sample of brain areas. Looking back at the tasks and the mental operations underlying them, we can conceive of the cognitive performance factor as a measurement of working memory, figural and spatial reasoning, text decoding, oral reading, auditory speech recognition and episodic memory. Various brain regions have been associated with these processes [125]. Particularly, large sections of the frontal cortex are associated with working memory [126,127]. Areas in the vicinity of the temporal and occipital cortex are related with language processing at different stages [128]. Spatial and figural reasoning have been reported to spread widely across the cortex [125,129,130]. The selection of areas we investigated does not cover all of these areas but is rather restricted. Additionally, the PM and the visual cortex process rather basic signals of perception and motor control instead of higher cognitive functions. In this regard, the finding of a nonexistent association between the widespread general factor of CT and general cognitive performance can be argued to align with previous literature, even though it contradicts our hypothesis 3. Although our findings do not directly contradict other studies reporting an association between the volume of the cortex layer and intelligence, because different brain regions were observed here, the association between brain and behaviour that did emerge is very small and restricted to temporal regions of the cortex, discouraging comparable endeavours in small samples.

Finally, the relationship between general cognitive performance and CT in the face-related network has to be interpreted heeding the functional meaning of the network specific CT factor. In the brain and behaviour models combining measures of performance and CT, CT was measured by combining all the vertices belonging to the respective parcels that seemed the most likely locations of the core areas of FC. However, the parcels probably also encompass regions not specialized for face processing. For instance, the ventral temporal lobe is also referred to as the ventral stream of visual processing, because projections from the visual cortex to these areas are associated with semantic visual processing, pattern recognition and speech processing [131–133]. In the HCP battery, speech and visual pattern processing tasks were available. It would be plausible that the relationship between general cognitive ability and CT in the face network is driven by complex visual processing. If this was the case, an alternative interpretation of the network specific CT factor would be that it represents complex visual processing instead of face processing.

## 6.4. On the impact of age, sex and handedness

We controlled for age, sex and handedness in our models using different methodological approaches. We discovered that even in this rather age-homogeneous sample of young adults, higher age was related with a thinner cortex when CT across the entire brain was taken into account, matching well-documented results of cortical thinning across adult age [134]. Relationships between CT in the face network and age were furthermore revealed, but only in the right hemisphere. Unexpectedly, the relationship between CT in the right face network and age was positive, contrary to the negative direction of the relationships between general CT and age. Thus, after the general negative relationship between CT and age was accounted for, the cortex in the face network was found to be thicker in older individuals.

In order to investigate sex differences in the model structures, we performed a multiple group analysis with sex as a grouping variable. Overall, the model structures were invariant across sexes, indicating that the individual differences found in the whole sample can be interpreted in the same way for male and female individuals. With respect to the brain and behaviour relationship, there was a numerical difference in lateralization between males and females, indicating that the relationship between face network CT and general performance was more right-sided in males and more left-sided in females, but the difference was not significant in this sample. Thus, it can be summarized that males and females showed a similar relationship between CT and performance. Additionally, the influence of sex on behavioural performance and CT was controlled by consistently including sex as a binary covariate into our measurement models. In summary, within this sample, males performed better than females when performance across all tasks was regarded, whereas females outperformed males in face-related tasks. The female superiority in terms of face-related performance is in line with

previous results obtained in FC paradigms [135]. Furthermore, females had overall thicker cortices in both hemispheres than males, a finding that has been reported before [136].

Finally, a small positive relationship between self-reported handedness and CT was detected, indicating that subjects who reported themselves to be more reliant on their right hand had slightly thicker cortices in the right-sided face network. However, when controlled for this relationship in our brain–behaviour models, no changes in the overall results emerged.

## 6.5. Limitations

Using a public dataset comes with several great advantages such as high data quality, high standardization of analysis pipelines and large sample sizes, to name some of the most obvious. However, there are downsides to using a protocol not composed with the specific purpose of investigating the variables of interest. In our case, particularly the interpretation of the face-related factor of performance is limited. With the data at hand, despite the high statistical power of the large sample, it is difficult to say whether the lack of relationship between face-related performance and face-related CT occurred because the hypothesis was incorrect, or because the face-related performance factor does not reflect true individual differences in a complex social ability such as FC. The same limitation applies to the localizer tasks. It is only natural that in a very extensive battery of tests such as the one applied in the HCP, every individual task has to be kept very parsimonious. However, as the lower reliability of the CT values in functionally localized areas shows, the localization of the core areas of FC was probably not optimal. Finally, but importantly, the results of the article are mainly based on analysis protocols that underwent *post hoc* adaptations, restricting interpretability.

## 7. Conclusion

With high statistical power, we demonstrated for the case of the core brain network of FC that areas of the cortex sharing a functional purpose did also share morphological properties in terms of CT. This is a novel finding with the potential to lead to a more comprehensive understanding of functional and developmental interrelation between brain areas. Thus, it carries importance for the concept of neural plasticity. However, more research is needed to investigate the factor structure of CT across the brain. Which other brain areas form interrelated networks in terms of morphological properties such as CT? If other latent variables of brain morphological features can be observed, what are their functional meanings? What is the biological mechanism underlying the common development of CT across different areas from a functionally related network? And importantly, how is the specific morphological development within a brain network related with behaviour and cognition? Particularly, the present results are mainly based on exploratory analyses that had to be adapted *post hoc* to registration. Thus, cross-validation in different datasets are needed. Nevertheless, in the present study, evidence from a large sample is provided that cognitive abilities can be predicted by CT as a property a brain network, in this case the core network of FC. More specialized studies with measurement paradigms tailored to the respective study aims are now needed in order to understand the role of latent variables of CT. Importantly, heed should be paid when composing tasks used for functional localization and for indicating domain-specific latent variables of behaviour. We showed that the present localizer tasks were not strong enough to yield reliable measures of CT, and that investigating larger ROIs is beneficial in this regard. Further, the size of the ROI strongly influenced the reliability of CT measurement. Based on these findings, we suggest that future studies use very strong localizer tasks or non-individual ROIs for investigating the morphology of face-selective brain regions. Furthermore, convergence across methods is needed to bring different brain measures together into more comprehensive models of neurological factor structure. How is network specific CT related, for instance, with latent factors of white matter integrity or functional connectivity? We encourage the pursuit of these highly relevant research questions in the future and emphasize that attempts of establishing SEMs with brain measures is a promising approach.

Ethics. The study conforms to Standard 8 of the American Psychological Association's Ethical Principles of Psychologist and Code of Conduct. Furthermore, the data are treated according to the WU-Minn HCP Consortium data use terms and the terms of use for the restricted data.

Data accessibility. The Stage 1 manuscript, unchanged from the date of in-principle acceptance (15/11/2018), is available at https://osf.io/p7c8z/. All data are provided by the open access HCP project mentioned above. Researchers

interested in future reanalysis for the purpose of reproduction and replication of the current results will have to acquire the data from the HCP repository directly (details on the neuroinformatics infrastructure have been provided by [137]). It should be considered that the HCP data protection policy permits the access to the demographic information by which individual subjects could potentially be identified only after every researcher involved in the study submitted an individual application to the HCP board to declare agreement with the HCP specific policy of data handling. These restricted-access data include the exact age and twin status of the subjects, among others. In order to control for sex, age and genetic relatedness of the studied individuals, we will acquire restricted data access, which will also be necessary for anyone aiming to pursue a similar research question by using the HCP data. We are not allowed to upload the HCP data in an open access portal. In contrast, all the analysis scripts that we developed will be shared with anyone interested. They will be provided along with a list of participants' codes of those subjects that we included in our analyses. To this purpose, we uploaded all materials to the third-party repository OSF. They can be accessed via the following link: https://osf.io/p7c8z/.

Authors' contributions. K.M. took part in designing the study, prepared the data analysis of behavioural as well as MRI data, wrote the first draft of the manuscript and addressed the reviewers' suggestions. After receiving in-principal acceptance, K.M. analysed the data according to the preregistered protocol and performed *post hoc* analysis steps. B.G. was involved in refining the study design, had a major part in preparing the preregistered and *post hoc* analysis of the MRI data, revised the draft of the manuscript and was involved in addressing the reviewers' suggestions. M.L. coordinated the analysis of the MRI data, provided expertise that enabled us to greatly improve the entire design including the hypotheses, had a major part in interpreting the results, revised the draft of the manuscript and helped address the reviewers' suggestions. A.H. conceived of the study, was involved in all the revisions of the study design, supervised the statistical analysis of the psychometric data (both preregistered and *post hoc*), helped to address revisions and was involved in interpreting the results and writing the manuscript. All of the authors approve of the article version that is submitted and ensure that questions related to the accuracy or integrity of any part of the work are, to our best knowledge, appropriately investigated and resolved.

Competing interests. We declare we have no competing interests.

Funding. Data were provided by the Human Connectome Project, WU-Minn Consortium (Principal Investigators: David Van Essen and Kamil Ugurbil; 1U54MH091657) funded by the 16 NIH Institutes and Centers that support the NIH Blueprint for Neuroscience Research; and by the McDonnell Center for Systems Neuroscience at Washington University. This research was further supported by a scholarship from the DAAD Germany granted to K.M. (ID: 57314604) to visit the Aging Research Center of the Karolinska Insitute and the Stockholm University to work with Prof. Martin Lövdén and Dr Benjamín Garzón.

Acknowledgements. Data were provided by the Human Connectome Project, WU-Minn Consortium (Principal Investigators: David Van Essen and Kamil Ugurbil; 1U54MH091657).

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
