## [Reviewer comments · Royal Society Open Science]

Review History

RSOS-180857.R0 (Original submission)

Review form: Reviewer 1 (Paul Thompson)

Is the language acceptable?

Yes

Do you have any ethical concerns with this paper?

No

Have you any concerns about statistical analyses in this paper?

Yes

Recommendation?

Major revision

Comments to the Author(s)

I have only reviewed the statistical elements of your manuscript. On the whole, your manuscript was well written and adopted best practice for reproducible research. I do, however, require some additional detail and justifications for some elements of your proposed statistical analyses. Please see corresponding document (Appendix A).

Many thanks,

Paul A. Thompson
University of Oxford

Review form: Reviewer 2**Is the language acceptable?**

Yes

Do you have any ethical concerns with this paper?

No

Have you any concerns about statistical analyses in this paper?

No

Recommendation?

Accept with minor revision

Comments to the Author(s)

RSOS-180857 - Brain Structure Associated with Accuracy of Face Cognition – Global and Specific Interindividual Differences in Cortical Thickness

1. The importance of the research question(s).

I think this research is important. There is strong behavioral evidence concerning the specificity of face and emotion processing in humans. There is also decent biological evidence concerning details of face and emotion processing. The interplay of both levels is somewhat less clearly established – specifically when it comes to cortical thickness.

2. The logic, rationale, and plausibility of the proposed hypotheses.

The logic of the study is straightforward. If face-perception and face-recognition are distinct abilities in the sense that they cannot be fully accounted for through other (established) abilities and if it is true that there are specialized regions for face-processing, then better performance should be associated with higher volume of specialized regions. The expectations are well motivated, the available data are strong in terms of allowing a sound test of the predictions, and the proposed analysis is well-suited to provide a strong conclusion.

3. The soundness and feasibility of the methodology and analysis pipeline (including statistical power analysis where applicable).

The study is based on a reanalysis of outstanding data. Obviously, power considerations might be deemed post hoc – but either way, given the large sample power is not a problem (unlike the situation in the majority of MRI studies). I have more expertise for the behavioral side of the study. The model proposed in figure 1 panel 2 stands a decent chance to fit the data (and better than the model in panel 1). Are the df in the model reported in the manuscript correct?

Obviously, if the same does not apply to figure 2 panels 2 and 1 respectively, there is little point in estimating the model in figure 3. Again, a nested factor for the behavioral side has been shown repeatedly – and similar evidence is presented for cortical thickness.

4. Whether the clarity and degree of methodological detail would be sufficient to replicate the proposed experimental procedures and analysis pipeline.

The procedure is really pretty clear. I am not an MRI expert but a quick search shows that there are alternative and arguably competing methods to measure/calculate/estimate cortical thickness. Here are some papers I found in a quick search:

<https://www.ncbi.nlm.nih.gov/pmc/articles/PMC2741580/>

<https://www.sciencedirect.com/science/article/pii/S1361841509000498>

https://www.researchgate.net/publication/51191142_A_comparison_of_voxel_and_surface_based_cortical_thickness_estimation_methods

https://www.researchgate.net/publication/5526073_Voxel-based_cortical_thickness_measurements_in_MRI

<https://www.sciencedirect.com/science/article/pii/S1053811916307637>

The authors should add a paragraph or two clarifying these issues.

The control of sex and age on the latent variable was not completely clear to me. Wouldn't it be better to partial sex and age effects on the level of manifest variables and move on analyzing residuals? At least for sex the sample size would permit estimating multiple group CFA – if that were an interesting research question. Given the available literature on sex differences in face-processing this might be the case.

5. Whether the authors provide a sufficiently clear and detailed description of the methods to prevent undisclosed flexibility in the experimental procedures or analysis pipeline.

The method is sufficiently clear and detailed. The “researcher degrees of freedom” are very limited. An independent team of researchers would be able to exactly replicate the models, results, and inferences.

Although I am not an expert, it might be an option to specify localized tasks differently.

Currently the authors propose to classify tasks according to stimulus content. An alternative could be to use only trials from 3.3.2.2 and ignore 3.3.2.1. I am not sure how decisive the nature of face stimuli is for completion of 3.3.2.1. May be there is evidence I am not aware of that shows that WM tasks with face content are face specific just like other face-tasks.

6. Whether the authors have considered sufficient outcome-neutral conditions (e.g. absence of floor or ceiling effects; positive controls; other quality checks) for ensuring that the results obtained are able to test the stated hypotheses.

Provided the available data allow specification of cortical thickness variables in non-face areas, these could be used to confirm that face specific relations are found for face specific behavioral and biological indicators but no such relationships can be found for non face specific data. This might strengthen the point the authors like to make.

Decision letter (RSOS-180857.R0)

31-Aug-2018

Dear Ms Meyer,

The Editors assigned to your Stage 1 Registered Report ("Brain Structure Associated with Accuracy of Face Cognition – Global and Specific Interindividual Differences in Cortical

Thickness") have now received comments from reviewers. We would like you to revise your paper in accordance with the referee and editors suggestions which can be found below (not including confidential reports to the Editor). Please note this decision does not guarantee eventual acceptance.

When submitting your revised manuscript, you must respond to the comments made by the referees and upload a file "Response to Referees" in "Section 2 - File Upload". Please use this to document how you have responded to the comments, and the adjustments you have made. In order to expedite the processing of the revised manuscript, please be as specific as possible in your response.

Kind regards,
Professor Chris Chambers
Royal Society Open Science
openscience@royalsociety.org

on behalf of Chris Chambers (Registered Reports Editor, Royal Society Open Science)
openscience@royalsociety.org

Associate Editor Comments to Author (Professor Chris Chambers):

Associate Editor: 1

Comments to the Author:

Two reviewers (including one statistical expert) have now appraised the manuscript. Both reviews are overall positive, but raise a number of specific issues that will be need to be addressed in order to achieve IPA. Among the main concerns, Reviewer 1 notes several areas of the statistical analysis where greater detail or justification is required (including, among other points, methods to deal with dependent observations, and the reporting of the power analysis), while Reviewer 2 asks for greater justification of the localizer tasks and suggestions for additional control analyses. Please attend carefully to each point raised in the reviews.

Comments to Author:

Reviewer: 1

Comments to the Author(s)

I have only reviewed the statistical elements of your manuscript. On the whole, your manuscript was well written and adopted best practice for reproducible research. I do, however, require some additional detail and justifications for some elements of your proposed statistical analyses. Please see corresponding document.

Many thanks,
Paul A. Thompson
University of Oxford

Reviewer: 2

Comments to the Author(s)

RSOS-180857 - Brain Structure Associated with Accuracy of Face Cognition – Global and Specific Interindividual Differences in Cortical Thickness

1. The importance of the research question(s).

I think this research is important. There is strong behavioral evidence concerning the specificity of face and emotion processing in humans. There is also decent biological evidence concerning details of face and emotion processing. The interplay of both levels is somewhat less clearly established – specifically when it comes to cortical thickness.

2. The logic, rationale, and plausibility of the proposed hypotheses.

The logic of the study is straightforward. If face-perception and face-recognition are distinct abilities in the sense that they cannot be fully accounted for through other (established) abilities and if it is true that there are specialized regions for face-processing, then better performance should be associated with higher volume of specialized regions. The expectations are well motivated, the available data are strong in terms of allowing a sound test of the predictions, and the proposed analysis is well-suited to provide a strong conclusion.

3. The soundness and feasibility of the methodology and analysis pipeline (including statistical power analysis where applicable).

The study is based on a reanalysis of outstanding data. Obviously, power considerations might be deemed post hoc – but either way, given the large sample power is not a problem (unlike the situation in the majority of MRI studies). I have more expertise for the behavioral side of the study. The model proposed in figure 1 panel 2 stands a decent chance to fit the data (and better than the model in panel 1). Are the df in the model reported in the manuscript correct? Obviously, if the same does not apply to figure 2 panels 2 and 1 respectively, there is little point in estimating the model in figure 3. Again, a nested factor for the behavioral side has been shown repeatedly – and similar evidence is presented for cortical thickness.

4. Whether the clarity and degree of methodological detail would be sufficient to replicate the proposed experimental procedures and analysis pipeline.

The procedure is really pretty clear. I am not an MRI expert but a quick search shows that there are alternative and arguably competing methods to measure/calculate/estimate cortical thickness. Here are some papers I found in a quick search:

<https://www.ncbi.nlm.nih.gov/pmc/articles/PMC2741580/>

<https://www.sciencedirect.com/science/article/pii/S1361841509000498>

https://www.researchgate.net/publication/51191142_A_comparison_of_voxel_and_surface_based_cortical_thickness_estimation_methods

https://www.researchgate.net/publication/5526073_Voxel-based_cortical_thickness_measurements_in_MRI

<https://www.sciencedirect.com/science/article/pii/S1053811916307637>

The authors should add a paragraph or two clarifying these issues.

The control of sex and age on the latent variable was not completely clear to me. Wouldn't it be better to partial sex and age effects on the level of manifest variables and move on analyzing residuals? At least for sex the sample size would permit estimating multiple group CFA – if that were an interesting research question. Given the available literature on sex differences in face-processing this might be the case.

5. Whether the authors provide a sufficiently clear and detailed description of the methods to prevent undisclosed flexibility in the experimental procedures or analysis pipeline. The method is sufficiently clear and detailed. The “researcher degrees of freedom” are very limited. An independent team of researchers would be able to exactly replicate the models, results, and inferences.

Although I am not an expert, it might be an option to specify localized tasks differently. Currently the authors propose to classify tasks according to stimulus content. An alternative could be to use only trials from 3.3.2.2 and ignore 3.3.2.1. I am not sure how decisive the nature of face stimuli is for completion of 3.3.2.1. May be there is evidence I am not aware of that shows that WM tasks with face content are face specific just like other face-tasks.

6. Whether the authors have considered sufficient outcome-neutral conditions (e.g. absence of floor or ceiling effects; positive controls; other quality checks) for ensuring that the results obtained are able to test the stated hypotheses.

Provided the available data allow specification of cortical thickness variables in non-face areas, these could be used to confirm that face specific relations are found for face specific behavioral and biological indicators but no such relationships can be found for non face specific data. This might strengthen the point the authors like to make.

Author's Response to Decision Letter for (RSOS-180857.R0)

See Appendix B.

RSOS-180857.R1 (Revision)

Review form: Reviewer 1 (Paul Thompson)

Is the language acceptable?

Yes

Do you have any ethical concerns with this paper?

No

Have you any concerns about statistical analyses in this paper?

No

Recommendation?

Accept in principle

Comments to the Author(s)

Thank you for addressing my comments and questions. I have no further comments and wish you all the best with your study.

Review form: Reviewer 2

Is the language acceptable?

Yes

Do you have any ethical concerns with this paper?

No

Have you any concerns about statistical analyses in this paper?

No

Recommendation?

Accept in principle

Comments to the Author(s)

The authors did a nice job addressing the first round of comments. I also like the comments of the other reviewer and the response provided by the authors. I therefore recommend to forward this paper to the 2nd stage.

Decision letter (RSOS-180857.R1)

15-Nov-2018

Dear Ms Meyer

On behalf of the Editor, I am pleased to inform you that your Manuscript RSOS-180857.R1 entitled "Brain Structure Associated with Accuracy of Face Cognition – Global and Specific Interindividual Differences in Cortical Thickness" has been accepted in principle for publication in Royal Society Open Science. The reviewers' and editors' comments are included at the end of this email.

You may now progress to Stage 2 and complete the study as approved. Before commencing data collection we ask that you:

- 1) Update the journal office as to the anticipated completion date of your study.
- 2) Please register your approved protocol, unchanged from the currently accepted version, on the Open Science Framework using the dedicated registration tool for RRs at the OSF (<https://osf.io/rr>), either publicly or privately under embargo until submission of the Stage 2 manuscript. Please note that a time-stamped, independent registration of the protocol is required under journal policy, and manuscripts that do not conform to this requirement cannot be considered at Stage 2. The protocol should be registered unchanged from its current approved state, with the time-stamp preceding implementation of the approved study design.

Following completion of your study, we invite you to resubmit your paper for peer review as a Stage 2 Registered Report. Please note that your manuscript can still be rejected for publication at Stage 2 if the Editors consider any of the following conditions to be met:

- The results were unable to test the authors' proposed hypotheses by failing to meet the approved outcome-neutral criteria.
- The authors altered the Introduction, rationale, or hypotheses, as approved in the Stage 1 submission.
- The authors failed to adhere closely to the registered experimental procedures. Please note that any deviations from the approved experimental procedures must be communicated to the editor immediately for approval, and prior to the completion of data collection. Failure to do so can result in revocation of in-principle acceptance and rejection at Stage 2 (see complete guidelines for further information).
- Any post-hoc (unregistered) analyses were either unjustified, insufficiently caveated, or overly dominant in shaping the authors' conclusions.
- The authors' conclusions were not justified given the data obtained.

We encourage you to read the complete guidelines for authors concerning Stage 2 submissions at <http://rsos.royalsocietypublishing.org/content/registered-reports>. Please especially note the requirements for data sharing, reporting the URL of the independently registered protocol, and that withdrawing your manuscript will result in publication of a Withdrawn Registration.

Please note that Royal Society Open Science will introduce article processing charges for all new submissions received from 1 January 2018. Registered Reports submitted and accepted after this date will ONLY be subject to a charge if they subsequently progress to and are accepted as Stage 2 Registered Reports. If your manuscript is submitted and accepted for publication after 1 January 2018 (i.e. as a full Stage 2 Registered Report), you will be asked to pay the article processing charge, unless you request a waiver and this is approved by Royal Society Publishing. You can find out more about the charges at <http://rsos.royalsocietypublishing.org/page/charges>. Should you have any queries, please contact openscience@royalsociety.org.

Once again, thank you for submitting your manuscript to Royal Society Open Science and we look forward to receiving your Stage 2 submission. If you have any questions at all, please do not hesitate to get in touch. We look forward to hearing from you shortly with the anticipated submission date for your stage two manuscript.

Kind regards,

Royal Society Open Science Editorial Office
Royal Society Open Science
openscience@royalsociety.org

on behalf of Professor Chris Chambers (Registered Reports Editor, Royal Society Open Science)
openscience@royalsociety.org

Reviewers' comments to Author:
Reviewer: 2

Comments to the Author(s)

The authors did a nice job addressing the first round of comments. I also like the comments of the other reviewer and the response provided by the authors. I therefore recommend to forward this paper to the 2nd stage.

Reviewer: 1

Comments to the Author(s)

Thank you for addressing my comments and questions. I have no further comments and wish you all the best with your study.

Author's Response to Decision Letter for (RSOS-180857.R1)

See Appendix C.

RSOS-180857.R2 (Revision)

Review form: Reviewer 2

Is the language acceptable?

Yes

Do you have any ethical concerns with this paper?

No

Have you any concerns about statistical analyses in this paper?

No

Recommendation?

Accept with minor revision

Comments to the Author(s)

Whether the data are able to test the authors' proposed hypotheses by passing the approved outcome-neutral criteria (such as absence of floor and ceiling effects or success of positive controls)

The data are really strong. The decisions the authors made in stage were smart and carefully considered. Nevertheless, analysis of the true data showed that some minor aspects of a priori settings were too strict (i.e. initial analysis with 200 subjects). It is nice, that the authors do not try to hide such issues. The same goes for the correlated errors on the behavioral side. They seem to make sense to me (and should have been prespecified). I cannot easily comment on the meaningfulness of the PM DLPFC correlation amongst residuals.

I am convinced for example that the analysis of competing methods for determining CT deserves a lot of attention in the field and the wisest decision could have been to publish analysis on this problem separately. This is probably not what should happen in the process of publishing a preregistered report. The analysis reported here shows, that using competing procedures for measuring CT delivers vastly different results. Not more than one method is best suited to measure CT. Therefore, for example, earlier results that are based on different methods are likely to deliver wrong results. Indeed, the present paper provides sobering results for supporters of CT-theories. The surprising and important result, that competing approaches for competing CT deliver vastly different results and are different in reliability should be elaborated. Such a section

for example should also include the magnitude of the relation of competing approaches to express CT across subjects.

Some of the important results should be moved from the supplement to the main text. After table 4 the authors should report associations of competing approaches to CT. Figures of CFA/SEM should include standardized parameter estimates or separate tables should be provided.

Given only the ROI mask procedure seems to work in the present data it doesn't strike me as essential to also report results from procedures that seemingly fail to achieve what they were designed for. Although it might not be in line with the policy of preregistration having reader go through text sections that later turn out as irrelevant because the underlying procedure is flawed seems somewhat unnecessary.

To be clear – I like the way data are handled here and I think over and beyond expected results the authors should also report upon the correlation of competing measures for measuring CT. I think the discussion should make some key results a little clearer. Agreed – the face factor wasn't very strong, but gathering stronger data together with brain data isn't strongly encouraged by the present data set. And the authors don't make this statement really clear. The same goes for the relation between CT and g. The conclusion drawn from the present data must be that the relation between the two is very small – despite the large sample size and statistical power non adherents of the theory that CT determines g would state it is zero. Given the authors use superior methods and data relative to earlier reports, odds are that earlier reports on a positive association are false positives. The present report should take a clear position concerning the consistency of different data sets.

Whether the Introduction, rationale and stated hypotheses are the same as the approved Stage 1 submission

Yes

Whether the authors adhered precisely to the registered experimental procedures
Sufficiently

Where applicable, whether any unregistered exploratory statistical analyses are justified, methodologically sound, and informative

Yes, absolutely.

Whether the authors' conclusions are justified given the data

Yes.

Decision letter (RSOS-180857.R2)

01-Jul-2019

Dear Ms Meyer:

On behalf of the Editor, I am pleased to inform you that your Stage 2 Registered Report RSOS-180857.R2 entitled "Are Global and Specific Interindividual Differences in Cortical Thickness Associated with Facets of Cognitive Abilities, Including Face Cognition?" has been deemed suitable for publication in Royal Society Open Science subject to minor revision in accordance with the referee suggestions. Please find the referees' comments at the end of this email.

The reviewer and Subject Editor have recommended publication, but also suggest some minor revisions to your manuscript. Therefore, I invite you to respond to the comments and revise your manuscript.

Please also ensure that all the below editorial sections are included where appropriate -- if any section is not applicable to your manuscript, please can we ask you to nevertheless include the heading, but explicitly state that the heading is inapplicable. An example of these sections is attached with this email.

- Ethics statement

- Data accessibility

If you wish to submit your supporting data or code to Dryad (<http://datadryad.org/>), or modify your current submission to dryad, please use the following link:
[http://datadryad.org/submit?journalID=RSOS&manu=\(Document not available\)](http://datadryad.org/submit?journalID=RSOS&manu=(Document not available))

- Competing interests

- Authors' contributions

- Acknowledgements

- Funding statement

Because the schedule for publication is very tight, it is a condition of publication that you submit the revised version of your manuscript within 7 days (i.e. by the 09-Jul-2019). If you do not think you will be able to meet this date please let me know immediately.

Please note that Royal Society Open Science will introduce article processing charges for all new submissions received from 1 January 2018. Registered Reports submitted and accepted after this date will ONLY be subject to a charge if they subsequently progress to and are accepted as Stage 2 Registered Reports. If your manuscript is submitted and accepted for publication after 1 January 2018 (i.e. as a full Stage 2 Registered Report), you will be asked to pay the article processing charge, unless you request a waiver and this is approved by Royal Society Publishing. You can find out more about the charges at <http://rsos.royalsocietypublishing.org/page/charges>. Should you have any queries, please contact openscience@royalsociety.org.

on behalf of Professor Chris Chambers (Editor)
 (Registered Reports Editor, Royal Society Open Science)
 openscience@royalsociety.org

Associate Editor Comments to Author (Professor Chris Chambers):

The Stage 2 manuscript was returned to one of the original Stage 1 reviewers (the other reviewer was not available), and I also read the manuscript myself. As you will see, the reviewer is overall very positive, which is a testament to the high quality of the study and judicious attention to detail throughout the submission. Nevertheless there are some relatively minor issues to address concerning the reporting and interpretation of results. In revising the manuscript to address the reviewer's comments, please take special care (a) not to change the introduction or framing of the study, and (b) not to allow the outcomes of exploratory post hoc analyses to dominate the interpretation of the results over the outcomes of the preregistered analyses, in either the Discussion or the Abstract. On an additional minor note, please ensure that the description of the Stage 1 manuscript is consistent across the various footnotes, e.g replacing "In the registered manuscript" in Footnote 6 with "In the accepted protocol..."

Concerning the reviewer's comment: "Although it might not be in line with the policy of preregistration having reader go through text sections that later turn out as irrelevant because the underlying procedure is flawed seems somewhat unnecessary." -- as the reviewer anticipates, it is important to maintain an accurate history of the methodological plan, therefore please do not remove this text.

Comments to Author:

Reviewer: 2

Comments to the Author(s)

Whether the data are able to test the authors' proposed hypotheses by passing the approved outcome-neutral criteria (such as absence of floor and ceiling effects or success of positive controls)

The data are really strong. The decisions the authors made in stage were smart and carefully considered. Nevertheless, analysis of the true data showed that some minor aspects of a priori settings were to strict (i.e. initial analysis with 200 subjects). It is nice, that the authors do not try to hide such issues. The same goes for the correlated errors on the behavioral side. They seem to make sense to me (and should have been prespecified). I cannot easily comment on the meaningfulness of the PM DLPFC correlation amongst residuals.

I am convinced for example that the analysis of competing methods for determining CT deserves a lot off attention in the field and the wisest decision could have been to publish analysis on this problem separately. This is probably not what should happen in the process of publishing a preregistered report. The analysis reported here show, that using competing procedures for measuring CT delivers vastly different results. Not more than one method is best suited to

measure CT. Therefore, for example, earlier results that are based on different methods are likely to deliver wrong results. Indeed, the present paper provides sobering results for supporters of CT-theories. The surprising and important result, that competing approaches for competing CT deliver vastly different results and are different in reliability should be elaborated. Such a section for example should also include the magnitude of the relation of competing approaches to express CT across subjects.

Some of the important results should be moved from the supplement to the main text. After table 4 the authors should report associations of competing approaches to CT. Figures of CFA/SEM should include standardized parameter estimates or separate tables should be provided.

Given only the ROI mask procedure seems to work in the present data it doesn't strike me as essential to also report results from procedures that seemingly fail to achieve what they were designed for. Although it might not be in line with the policy of preregistration having reader go through text sections that later turn out as irrelevant because the underlying procedure is flawed seems somewhat unnecessary.

To be clear – I like the way data are handled here and I think over and beyond expected results the authors should also report upon the correlation of competing measures for measuring CT. I think the discussion should make some key results a little clearer. Agreed – the face factor wasn't very strong, but gathering stronger data together with brain data isn't strongly encouraged by the present data set. And the authors don't make this statement really clear. The same goes for the relation between CT and g. The conclusion drawn from the present data must be that the relation between the two is very small – despite the large sample size and statistical power non adherents of the theory that CT determines g would state it is zero. Given the authors use superior methods and data relative to earlier reports, odds are that earlier reports on a positive association are false positives. The present report should take a clear position concerning the consistency of different data sets.

Whether the Introduction, rationale and stated hypotheses are the same as the approved Stage 1 submission

Yes

Whether the authors adhered precisely to the registered experimental procedures

Sufficiently

Where applicable, whether any unregistered exploratory statistical analyses are justified, methodologically sound, and informative

Yes, absolutely.

Whether the authors' conclusions are justified given the data

Yes.

Author's Response to Decision Letter for (RSOS-180857.R2)

See Appendix D.

Decision letter (RSOS-180857.R3)

10-Jul-2019

Dear Ms Meyer:

It is a pleasure to accept your Stage 2 Registered Report entitled "Are Global and Specific Interindividual Differences in Cortical Thickness Associated with Facets of Cognitive Abilities, Including Face Cognition?" in its current form for publication in Royal Society Open Science.

on behalf of Professor Chris Chambers (Subject Editor)
openscience@royalsociety.org

Appendix A

Article: Brain Structure Associated with Accuracy of Face Cognition – Global and Specific Interindividual Differences in Cortical Thickness

Authors: Meyer, Kristina; Garzón, Benjamín; Lövdén, Martin; Hildebrandt, Andrea;

Reviewer: Paul A. Thompson, University of Oxford.

Overview:

The article discusses a proposed investigation of the relationship between Cortical thickness (a neural correlate of performance and learning) in the brain network of Facial cognition and performance on a battery of psychometric tests, specifically using tasks with facial content.

The investigation proposes to use structural equation modelling to quantify the relationships.

Review limitation:

I will only comment on the statistical elements of the article.

1. The importance of the research question(s).

I am not a subject expert in this field, so I can only give my opinion based on the information presented. The research question appears important given that there has been limited previous work relating individual differences in facial recognition to brain structure.

2. The logic, rationale, and plausibility of the proposed hypotheses.

The discussion of the proposed hypotheses is well explained and shows good judgement to build the model incrementally (Hypothesis 1 -> Hypothesis 3).

The authors have been transparent when discussing the open access data and avoiding generation of the hypotheses using a sub-sample of the data. I think this is a valid approach and the authors correctly discard this sub-sample from the main analysis. The sub-sample appears to be a reasonable size for the purpose of exploratory analysis to determine psychometric quality of the data and give insight for power calculation simulations.

3. The soundness and feasibility of the methodology and analysis pipeline (including statistical power analysis where applicable).

I will discuss the details of the methods in the order that they appear in the article.

Priori power analysis – The authors use Monte Carlo simulation to justify power, (which would also be my choice) and is the gold standard in SEM. I have some concerns that the estimate of effect size for Hypothesis 1 relies on sparse information that is, probably, of variable quality given the sample size of previous studies discussed earlier in the manuscript. In particular, the justification for nested face factors was too vague.

I would prefer that the authors give an improved justification perhaps looking at power for a range of expected effect sizes that would be within a meaningful window of values. I do not suggest this to be difficult as the authors have clearly made effort to estimate this accurately and I am happy to be convinced by the authors in rebuttal, why their justification is adequate.

With regard to hypothesis 2 and 3, the power has again been determined via simulation using estimates from the preliminary analysis. I think this approach is reasonable given the size of the sub-sample for rough estimation of the effect sizes. Had the sub-sample been smaller, I would have been considerably more critical. Pilot data for estimation of parameters is a typical method, although there is no guarantee that the estimates will be reflective of the parameters in the main data. I feel they will be sufficient for power calculations in this instance.

Handling Behavioural data and preliminary analyses – I think the authors are correct here in their approach to use of preliminary analyses. If the main analyses were to be conducted without consideration of data properties, such as ceiling effects and normality of distributions, then it would be highly likely that the simulations for power would be misleading and potentially lead to model fit issues and type I error as the authors hint. I am also pleased to see that the authors report that they will use cross validation in the main analysis to add a degree of robustness to their findings.

The authors report the SEM models correctly and conclusions drawn in this section are accurate. The addition of the accF factor shows a clear improvement of model fit and it will be interesting to see if the findings are replicated in the main data analysis. I agree that the factor loadings should be constrained for identical tasks at different time points as is standard practice.

MRI data processing and analysis – Not much to comment on here. The procedures are generally contained within standard software and the authors give reference to versions and pre-processing details. The only deviation is use of some custom scripts which the authors state “will be made available via OSF upon publication” which is encouraging for reproducibility.

Dealing with Lateralization – I am not an expert in lateralization, so the author’s prediction seems reasonable given the previously cited literature. Specifically documenting the right hemisphere expectation of results is worth noting. I wonder if the authors have more specific predictions of differences in model structure depending on side of ROI (this is purely out of interest rather than a necessary change – authors can feel free to address this or not).

Dealing with potential confounds through handedness and family relationships – I am happy with the control of handedness, as I believe this is appropriate. However, I would like much more detail from the authors on the family relationships as this is a highly significant part of the model. Why do the authors avoid using a multilevel approach to deal with the dependent observations? This needs full justification including references to their proposed approach.

ROI extraction – Could the authors comment on whether they believe that depending on how the parcelling of smaller groups into larger sections occurs, does this alters the SEM model results?

Test-Retest Reliability of CT measures – Fine.

SEM – Standard procedures documented here and entirely appropriate for the planned analysis.

4. Whether the clarity and degree of methodological detail would be sufficient to replicate the proposed experimental procedures and analysis pipeline.

SEM – The analyses are generally well-documented, but I would like further clarity on the “Dealing with potential confounds” specifically for family relationships. All R and Mplus scripts will be documented and made available on an appropriate repository. In particular, scripts for power should be uploaded prior to formal analysis.

MRI data processing – all documented adequately and references to version and procedures provided. Additional custom scripts will be made available via OSF.

ROI extraction – The method seems appropriate but I would like to see that the model result is not dependent on ROI specification. I do not know enough about these methods to say whether the ROIs can be pre-specified before main SEM analysis?

5. Whether the authors provide a sufficiently clear and detailed description of the methods to prevent undisclosed flexibility in the experimental procedures or analysis pipeline.

See 4.

6. Whether the authors have considered sufficient outcome-neutral conditions (e.g. absence of floor or ceiling effects; positive controls; other quality checks) for ensuring that the results obtained are able to test the stated hypotheses.

I think that the authors have considered this and made efforts to ensure an outcome-neutral result. This was achieved using a sub-sample investigation.

Appendix B

Prof. Dr. Chris Chambers, Editor
Royal Society Open Science
Cardiff University
CUBRICK, Maindy Road
Cardiff CF24 4HQ

Oldenburg, October 29th, 2018

Dear Professor Chris Chambers,

Hereby, we resubmit the manuscript “Brain Structure Associated with Accuracy of Face Cognition – Global and Specific Interindividual Differences in Cortical Thickness” for further evaluation as a Registered Report in the Journal “Royal Society Open Science” (RSOS).

We thank you and the reviewers for granting us the opportunity for this revision of our registered report at stage 1. We were happy to read that the reviewers agreed with us in several points and expressed an overall positive evaluation of our approach. In the following, you will find our numbered replies describing in which ways we addressed the comments and suggestions the reviewers expressed. We are convinced that the study will greatly benefit from these inspiring ideas.

The resubmitted version of the article contains revisions that are highlighted in dark red color.

Comments to the authors by Reviewer #1:

Review limitation: I will only comment on the statistical elements of the article.

The importance of the research question(s).

I am not a subject expert in this field, so I can only give my opinion based on the information presented. The research question appears important given that there has been limited previous work relating individual differences in facial recognition to brain structure.

The logic, rationale, and plausibility of the proposed hypotheses.

The discussion of the proposed hypotheses is well explained and shows good judgement to build the model incrementally (Hypothesis 1 -> Hypothesis 3). The authors have been transparent when discussing the open access data and avoiding generation of the hypotheses using a sub-sample of the data. I think this is a valid approach and the authors correctly discard this sub-sample from the main analysis. The sub-sample appears to be a reasonable size for the purpose of exploratory analysis to determine psychometric quality of the data and give insight for power calculation simulations.

The soundness and feasibility of the methodology and analysis pipeline (including statistical power analysis where applicable).

I will discuss the details of the methods in the order that they appear in the article.

[1] Priori power analysis – The authors use Monte Carlo simulation to justify power, (which would also be my choice) and is the gold standard in SEM. I have

some concerns that the estimate of effect size for Hypothesis 1 relies on sparse information that is, probably, of variable quality given the sample size of previous studies discussed earlier in the manuscript. In particular, the justification for nested face factors was too vague. I would prefer that the authors give an improved justification perhaps looking at power for a range of expected effect sizes that would be within a meaningful window of values. I do not suggest this to be difficult as the authors have clearly made effort to estimate this accurately and I am happy to be convinced by the authors in rebuttal, why their justification is adequate.

We agree that effect sizes reported in other studies are not exhaustive estimators of the effects to be expected in our prospective sample. Although intuitively we think that the sample is convincingly large (especially as compared with many other neuroscience studies in the field), we now apply a range of effect sizes in the Monte Carlo power simulation as the reviewer suggested. Note, however, that the loadings on the nested factor were already set very conservatively. We assume standardized factor loadings of .60 on the general factor of cortical thickness and .30 on the nested face-specific factor of cortical thickness, taking into account that loadings on nested factors are typically lower. Assuming standardized loadings below .30 with the given number of indicators will clearly lead to problems identifying the nested factor. We now use a range of +/- .10 around the factor loadings on the general factor, but we see .30 as the lowest possible value to identify the nested factor. The power simulation including a range around the effect sizes (factor loadings) and a few sentences on the matter of loadings lower than .30 have been added to the manuscript (p. 4).

With regard to hypothesis 2 and 3, the power has again been determined via simulation using estimates from the preliminary analysis. I think this approach is reasonable given the size of the subsample for rough estimation of the effect sizes. Had the sub-sample been smaller, I would have been considerably more critical. Pilot data for estimation of parameters is a typical method, although there is no guarantee that the estimates will be reflective of the parameters in the main data. I feel they will be sufficient for power calculations in this instance.

Handling Behavioural data and preliminary analyses – I think the authors are correct here in their approach to use of preliminary analyses. If the main analyses were to be conducted without consideration of data properties, such as ceiling effects and normality of distributions, then it would be highly likely that the simulations for power would be misleading and potentially lead to model fit issues and type I error as the authors hint. I am also pleased to see that the authors report that they will use cross validation in the main analysis to add a degree of robustness to their findings.

The authors report the SEM models correctly and conclusions drawn in this section are accurate. The addition of the accF factor shows a clear improvement of model fit and it will be interesting to see if the findings are replicated in the main data analysis. I agree that the factor loadings should be constrained for identical tasks at different time points as is standard practice.

MRI data processing and analysis – Not much to comment on here. The procedures are generally contained within standard software and the authors give reference to versions and pre-processing details. The only deviation is use of some custom scripts

which the authors state “will be made available via OSF upon publication” which is encouraging for reproducibility.

[2] Dealing with Lateralization – I am not an expert in lateralization, so the author’s prediction seems reasonable given the previously cited literature. Specifically documenting the right hemisphere expectation of results is worth noting. I wonder if the authors have more specific predictions of differences in model structure depending on side of ROI (this is purely out of interest rather than a necessary change – authors can feel free to address this or not).

As we also point out in the paper, there is some evidence that face-responsive brain areas in the right hemisphere show a stronger activation than the respective right-sided areas when subjects view face stimuli. However, to our knowledge, there is no indication in the literature letting us believe that the postulated model structure of individual differences in cortical thickness (CT) will be hemisphere-specific. In line with the literature on mean differences in activation strength between the hemispheres, we expect the intercepts of indicators originating from the CT measures in the left hemisphere to be lower. Factor loadings may or may not be lower, because the magnitude of loadings only partly depends on the mean structure through potential variance restriction. If factor loadings of indicators from the left hemisphere are constricted (lower factor reliability) the prediction accuracy of face cognition performance by a specific nested factor of CT in the left hemisphere would be expected to be somewhat weaker. We do however not expect the relations for the left hemisphere indicators to be absent or reversed, but we now plan to explore any arising effect differences outside the postulated boundaries in an exploratory part of our study (this is because precise predictions cannot be made in the light of the present literature).

We added a sentence in the manuscript (p. 8) to explain this.

[3] Dealing with potential confounds through handedness and family relationships – I am happy with the control of handedness, as I believe this is appropriate. However, I would like much more detail from the authors on the family relationships as this is a highly significant part of the model. Why do the authors avoid using a multilevel approach to deal with the dependent observations? This needs full justification including references to their proposed approach.

We fully agree that nestedness of the data due to family relations needs to be accounted for. A multilevel modeling approach is the way to deal with nested (clustered) data structures, but this method would be justified if we had research questions concerning the clusters. Instead, we are interested in how the relations generally vary across individuals, independently on whether the within cluster relations vary across family clusters. In the HCP data we have the challenge that some observations are not independent, but clustered based on family relatedness – this concerns only a part of the sample. There are statistical approaches that allow to correct for dependency of observations. Given that our research questions are related to overall relationships we view an approach where we control/correct for nestedness as the most suitable approach. Additionally, such an approach is appropriate, because only a part of the sample includes persons in family relationships. MPlus offers a solution for correcting standard error bias that will be induced through the dependency of observations. The ANALYSIS=COMPLEX command, after specifying the family dependency as a clustering variable, will provide

corrected standard errors without estimating within and between cluster coefficients. Thus, with this option, standard errors and the chi-square test of model fit are adjusted, taking into account non-independence of observations due to clustered sampling (see Muthén & Muthén, 1998-2010, pp. 239 ff). Simulation studies show that this method is well suited to adjust for interdependence of observations (Muthén & Satorra, 1995). The method has been applied and evaluated in further studies (Asparouhov, 2005, 2006; Asparouhov & Muthén, 2005, 2006). The general recommendation is to use this option if there are at least 20 clusters, dependent observations. The HCP sample comprises 456 families of varying configuration. We realize that this choice was not communicated clearly enough in the previous version of our manuscript. We thus added a few sentences to the section 4.3.2 (p. 8) for clarification.

A further alternative proposed in the literature would be the use of a permutation method published by Winkler et al. (2015). This approach deals with interdependence of observations by introducing blocks of observations for which restrictions in exchangeability are set. Observations belonging to a cluster, such as a family, cannot be shuffled freely, but only within boundaries with respect to their family status. This approach has been applied to a part of the HCP data in Winkler et al. (2015) and it was found to perform well at investigating heritability in the sample. Because we do not aim to address questions with respect to heritability, we prefer to preregister the aforementioned and well-established method by Muthén and Satorra (1995) that corrects standard errors biased through observations' dependency.

Muthén, L.K. and Muthén, B.O. (1998-2010). *Mplus User's Guide*. Sixth Edition. Los Angeles, CA: Muthén & Muthén

Muthen, B. O., & Satorra, A. (1995). Complex sample data in structural equation modeling. *Sociological methodology*, 267-316.

Winkler, A. M., Webster, M. A., Vidaurre, D., Nichols, T. E., & Smith, S. M. (2015). Multi-level block permutation. *Neuroimage*, 123, 253-268.

[4] ROI extraction – Could the authors comment on whether they believe that depending on how the parcelling of smaller groups into larger sections occurs, does this alters the SEM model results?

Thank you for this question. We understand and agree with your concern that the use of different parcels may lead to somewhat different results.

The use of broader and more wide-spread sections of the cortex is relevant for the present study. This way, we hope to capture variance in CT within a person across the brain. These larger sections have been found by Glasser and colleagues (2016), who used functional, structural and cytoarchitectonic information in order to differentiate brain areas from each other. The larger sections can be broken down into smaller parcels. In the end, the calculation in our case will be as follows: We will consider the entire selected sections, leave out those parcels which we know from previous research to be related with face processing, and compute the average CT from all the vertices comprised in each of the parcels in that section. The ROIs used to indicate the specific face-related factor are much smaller than that, but it is our aim to cover a larger section of the brain for

indicating a general CT factor. Importantly, the choice of the ROI for indicating the general CT factor will not be data-driven. It is decided which parcels we will use. Our exact choices of parcels corresponding to the parcellation of Glasser and colleagues (2016) and they are exactly specified in our scripts. Thus, there will be no data driven researchers' degrees of freedom involved in this choice.

We added a few sentences to the manuscript (p. 9) to communicate this approach more clearly.

Test-Retest Reliability of CT measures – Fine.

SEM – Standard procedures documented here and entirely appropriate for the planned analysis.

Whether the clarity and degree of methodological detail would be sufficient to replicate the proposed experimental procedures and analysis pipeline.

SEM – The analyses are generally well-documented, but I would like further clarity on the “Dealing with potential confounds” specifically for family relationships.

All R and Mplus scripts will be documented and made available on an appropriate repository. In particular, scripts for power should be uploaded prior to formal analysis.

MRI data processing – all documented adequately and references to version and procedures provided. Additional custom scripts will be made available via OSF.

[5] ROI extraction – The method seems appropriate but I would like to see that the model result is not dependent on ROI specification. I do not know enough about these methods to say whether the ROIs can be pre-specified before main SEM analysis?

We agree that it is reasonable to pre-specify the ROIs before the SEM analysis (see issue #[4]). Because it seemed too arbitrary to parcel the entire cortex ourselves, we chose the Glasser et al. (2016) parcellation as a basis. This modern atlas segregates the brain into 22 large regions which in turn can be broken down into a total of 180 smaller regions. Importantly, the parcellation is based on several structural and functional measures, but was created without relation to the research questions asked here.

We applied a number of criteria to choose the most appropriate ROIs for later analyses. Mostly, the reliability of CT measures found in other studies and the fact whether or not a section of the brain includes a face-related area were taken into account. Importantly, because we agree that the choice of too narrow ROIs for the general factor of CT would probably alter the results, we picked regions that spread widely across the entire cortex and are functionally not strongly related with each other. This way, we aim to avoid that the regions share much variance due to a common functionality, as we expect the face-selective regions to do.

Whether the authors provide a sufficiently clear and detailed description of the methods to prevent undisclosed flexibility in the experimental procedures or analysis pipeline.

See above.

Whether the authors have considered sufficient outcome-neutral conditions (e.g. absence of floor or ceiling effects; positive controls; other quality checks) for ensuring that the results obtained are able to test the stated hypotheses.

I think that the authors have considered this and made efforts to ensure an outcome-neutral result. This was achieved using a sub-sample investigation.

Comments to the authors by Reviewer #2

The importance of the research question(s).

I think this research is important. There is strong behavioral evidence concerning the specificity of face and emotion processing in humans. There is also decent biological evidence concerning details of face and emotion processing. The interplay of both levels is somewhat less clearly established – specifically when it comes to cortical thickness.

The logic, rationale, and plausibility of the proposed hypotheses.

The logic of the study is straightforward. If face-perception and face-recognition are distinct abilities in the sense that they cannot be fully accounted for through other (established) abilities and if it is true that there are specialized regions for face-processing, then better performance should be associated with higher volume of specialized regions. The expectations are well motivated, the available data are strong in terms of allowing a sound test of the predictions, and the proposed analysis is well-suited to provide a strong conclusion.

The soundness and feasibility of the methodology and analysis pipeline (including statistical power analysis where applicable).

The study is based on a reanalysis of outstanding data. Obviously, power considerations might be deemed post hoc – but either way, given the large sample power is not a problem (unlike the situation in the majority of MRI studies).

[6] I have more expertise for the behavioral side of the study. The model proposed in figure 1 panel 2 stands a decent chance to fit the data (and better than the model in panel 1). Are the df in the model reported in the manuscript correct? Obviously, if the same does not apply to figure 2 panels 2 and 1 respectively, there is little point in estimating the model in figure 3.

Thank you for pointing to the typo with respect to the degrees of freedom. We corrected them (p. 6). Furthermore, we agree that the model in figure 3 only needs to be estimated if the nested face-specific factor of CT turns out to be identified. This is the case if the CT model including the nested factor would fit the data better than the general CT factor model only. We added a sentence describing this detail of the model sequence to the sections 4.4 (p. 8) and 4.4.3 (p. 9), respectively.

Again, a nested factor for the behavioral side has been shown repeatedly – and similar evidence is presented for cortical thickness.

Whether the clarity and degree of methodological detail would be sufficient to replicate the proposed experimental procedures and analysis pipeline.

[7] The procedure is really pretty clear. I am not an MRI expert but a quick search shows that there are alternative and arguably competing methods to measure/calculate/estimate cortical thickness. Here are some papers I found in a quick search:

<https://www.ncbi.nlm.nih.gov/pmc/articles/PMC2741580/>

<https://www.sciencedirect.com/science/article/pii/S1361841509000498>

https://www.researchgate.net/publication/51191142_A_comparison_of_voxel_and_surface_based_cortical_thickness_estimation_methods

https://www.researchgate.net/publication/5526073_Voxel-based_cortical_thickness_measurements_in_MRI

<https://www.sciencedirect.com/science/article/pii/S1053811916307637>

The authors should add a paragraph or two clarifying these issues.

We added a paragraph to the Introduction section of the manuscript (p. 2) and the Methods section (p. 4) explaining why we favor surface-based over voxel-based morphometry (VBM) approaches. Mostly, VBM methods appear difficult to interpret and less reliable in comparison to SBM methods. SBM methods, in turn, yield measures such as the geometrically measurable thickness of the outer cortex, which directly represents an anatomical feature of the brain. This seems more plausible and straight-forward to interpret.

In addition to this general preference grounded in theoretical reasons, there is also a practical advantage to use CT instead of VBM measures in the present study: The HCP developers provide CT maps of the brain which have been obtained in collaboration with the software developers of FreeSurfer and FSL. These maps have been created using state-of-the-art statistical methods that are designed and optimized specifically for dealing with the high dimensional HCP data. Building upon and trusting the extensive expertise of the HCP researchers, we do not anticipate to come up with an equally good or even better pipeline ourselves, should we choose to use VBM measures instead. Thus, for practical as well as for theoretical reasons, we decided to stick to the established alternative by the HCP experts. Using these measures reduces costs in terms of time and fosters reproducibility by other researcher.

[8] The control of sex and age on the latent variable was not completely clear to me. Wouldn't it be better to partial sex and age effects on the level of manifest variables and move on analyzing residuals? At least for sex the sample size would permit estimating multiple group CFA – if that were an interesting research question. Given the available literature on sex differences in face-processing this might be the case.

Thanks for your suggestions.

Age: Given that all subjects are younger adults, we expect that there will be no considerable effects of age in the sample. Therefore, we think that our approach – keeping age as a covariate – should be sufficiently prudent and well suited to deal with potential, but expectedly minor age effects. Because our main hypotheses are related to parameters at the latent level, we aim to control for age at the level of the latent variables.

Sex: We agree that additionally performing the analyses simultaneously on two groups would be interesting to investigate how sex would moderate the findings. Above behavioral findings on sex differences mentioned by the reviewer, there is also research showing that the lateralization of activity in face-responsive areas depends on sex, meaning that it was found in males, but not in females:

Proverbio, A. M., Brignone, V., Matarazzo, S., Del Zotto, M., & Zani, A. (2006). Gender differences in hemispheric asymmetry for face processing. *BMC neuroscience*, 7(1), 44.

Therefore, we will perform an additional separate exploratory analysis investigating sex differences. The procedure is now described in the methods section (p. 8).

Whether the authors provide a sufficiently clear and detailed description of the methods to prevent undisclosed flexibility in the experimental procedures or analysis pipeline.

The method is sufficiently clear and detailed. The “researcher degrees of freedom” are very limited. An independent team of researchers would be able to exactly replicate the models, results, and inferences.

[9] Although I am not an expert, it might be an option to specify localized tasks differently. Currently the authors propose to classify tasks according to stimulus content. An alternative could be to use only trials from 3.3.2.2 and ignore 3.3.2.1. I am not sure how decisive the nature of face stimuli is for completion of 3.3.2.1. May be there is evidence I am not aware of that shows that WM tasks with face content are face specific just like other face-tasks.

In Barch et al. (2013), the localizer tasks are described and tested. The authors explain that if the research aim is to identify cortex regions responding to specific stimulus types, “one can collapse across memory load and focus only on stimulus type comparisons“ (p. 172). This is the case for our study. In other studies, WM tasks are used as localizer tasks for identifying face-responsive areas as well (e.g. Ishai, Pessoa, Bikle, & Ungerleider, 2004; Rossion et al., 2003). The localizer tasks in the HCP study are relatively easy and face stimuli can be contrasted with non-face stimuli. Altogether, we believe that the procedure localizing in terms of content is very common, which is also the reason why these two tasks using facial stimuli were included in the HCP measurement battery.

Barch, D. M., Burgess, G. C., Harms, M. P., Petersen, S. E., Schlaggar, B. L., Corbetta, M., ... & Nolan, D. (2013). Function in the human connectome: task-fMRI and individual differences in behavior. *Neuroimage*, 80, 169-189.

Ishai, A., Pessoa, L., Bikle, P. C., & Ungerleider, L. G. (2004). Repetition suppression of faces is modulated by emotion. *Proceedings of the National Academy of Sciences*, 101(26), 9827-9832.

Rossion, B., Caldara, R., Seghier, M., Schuller, A. M., Lazeyras, F., & Mayer, E. (2003). A network of occipito- temporal face- sensitive areas besides the right middle fusiform gyrus is necessary for normal face processing. *Brain*, 126(11), 2381-2395.

We agree that alternatively conducting a process based localization to be contrasted with stimulus content localization in terms of resulting relationships with psychometric

performance would be an interesting research question on its own. We however think that it would go beyond the present scope and decided to keep the aims straightforward and conduct a stimulus type based localization here. We thank the reviewer for the inspiring idea which we consider very interesting for future research.

Whether the authors have considered sufficient outcome-neutral conditions (e.g. absence of floor or ceiling effects; positive controls; other quality checks) for ensuring that the results obtained are able to test the stated hypotheses.

[10] Provided the available data allow specification of cortical thickness variables in non-face areas, these could be used to confirm that face specific relations are found for face specific behavioral and biological indicators but no such relationships can be found for non face specific data. This might strengthen the point the authors like to make.

We agree that it would be interesting to introduce additional nested factors, encompassing CT in other specific, but not face-related areas of the brain. However, we see several disadvantages in such an approach. First and foremost, there are specific hypotheses as to which areas should belong to the core network of face cognition. This makes it relatively easy to predict which areas might share variance in their CT if, as we assume, the use of these areas in simultaneous tasks leads to a simultaneous growth of CT in these areas. We can imagine that such common variance might be found among other, functionally related brain regions as well. However, it currently seems arbitrary which areas would be well suited for such an endeavor. Second, the entire analysis sequence will be quite extensive already. Adding additional nested factors for object/place processing for example would take away some of its parsimony and render the results to become more difficult to interpret. Finally, results will be meaningful with respect to methodology of structural MRI, but the approach is quite new. We would think it preferable to test the approach in a circumscribed number of ROIs and potentially extend and refine it later by adding other factors for comparison, if the approach turns out to be promising. We will discuss this future potential when we are submitting our full manuscript including results and discussion.

Again, we express our sincere gratitude for your constructive feedback. We hope that our revisions erase the reviewer's concerns and that we will soon be able to start the data analysis.

Sincerely,

Kristina Meyer, Benjamín Garzón, Martin Lövdén, Andrea Hildebrandt

Appendix C

Prof. Dr. Chris Chambers, Editor
Royal Society Open Science
Cardiff University
CUBRICK, Maindy Road
Cardiff CF24 4HQ

Oldenburg, June 5th, 2019

Dear Professor Chris Chambers, dear editors,

We hereby resubmit the registered report RSOS-180857.R2 for a second review (stage 2). The manuscript was unsubmitted on June 3rd to give us the chance to address suggestions offered by the editorial office. We were thankful to receive your sensible comments.

In the study, we investigated whether we can predict specific face cognition abilities by individual differences in cortical thickness (CT) in the core brain network of face cognition. Using structural equation modeling (SEM), we discovered a latent factor architecture of CT reminiscent of models explaining behavioral performance. This is evidence that in line with our first hypothesis, brain areas sharing the purpose of processing faces develop morphologically in parallel. When investigating the relationship between brain and behavior, however, a pattern was discovered that was not predicted. We expected CT in the core network of face cognition to predict performance in tasks with face stimuli. Instead, CT in the face network predicted general cognitive performance.

Thank you for providing guidance to correct errors found in the accepted part of the manuscript in your response we received on June 3rd. As suggested, we (1) ensured that the link to the stage 1 manuscript is now more explicitly mentioned on page 22 in the Data Accessibility section, we (2) changed the introduction and methods section to the past tense and (3) altered the title and summary in accordance with the study conclusion. Finally, we made necessary corrections to the preregistered part of the protocol more visible to the readers by adding footnotes and a specific section in the manuscript right before the Results section (4.5 Deviations from Protocol). Furthermore, changes in the manuscript can still be tracked to make them easily detectable for the reviewers. If you or the reviewers are of a different opinion in some of the points, we would be grateful if you could get in contact with us. It is our express desire to be fully transparent and support this type of publication, but we also think that no paper can be published while we know it contains inconsistencies.

We confirm that we applied the preregistered analysis in its approved form. Furthermore, the analysis was not started before receiving in-principal-acceptance from

RSOS. Any preliminary analyses were approved at stage 1 and included in the stage 1 manuscript. All data are freely available after registration at the Human Connectome Project (www.humanconnectome.org). All scripts and subject IDs necessary to repeat our analyses are available on our study's OSF site (the link can be found on the manuscript pages 9, 17 and 22): <https://osf.io/p7c8z/>

After adhering to our preregistered analysis protocol, we discovered that not all of our decisions we made in the stage 1 protocol were optimal. Despite our efforts to perform a preliminary analysis in a sub-sample of $N = 200$ participants, we found that the models we preregistered did only partly yield interpretable results when applied to the full sample of $N = 854$. In SEM, a good model fit is a crucial prerequisite for drawing any meaningful conclusions about the parameter estimates. Thus, in addition to reporting the results of the preregistered analysis without any changes, we added post-hoc analyses that are clearly marked as such in the manuscript. In these, we altered the SEMs in ways that better accounted for the data. These changes did not alter the logic of the model sequences and were rather minor. Furthermore, we discovered unusually low manifest correlations between some of the CT measures. Thus, alongside our preregistered reliability analysis of CT measurements, we provided an additional comparison between three different ways of defining face-related brain regions of interest (ROIs).

Again, we confirm that the study conforms to Standard 8 of the American Psychological Association's Ethical Principles of Psychologist and Code of Conduct. The data were treated according to the WU-Minn HCP Consortium data use terms and the terms of use for the restricted data. The authors declare no conflict of interest. The manuscript does not include any copyrighted material. The present manuscript is an original work, it is not published or being considered for publication anywhere else (partially or entirely).

We thank you once more for your kind support and guidance. We look forward to hearing from you.

Sincerely,

Kristina Meyer, Benjamín Garzón, Martin Lövdén, Andrea Hildebrandt

1: Carl von Ossietzky Universität Oldenburg
Department of Psychology
Ammerländer Heerstraße 114-118
26129 Oldenburg, Germany

kristina.meyer@uni-oldenburg.de

Telephone +49 (0)441 798-4578

andrea.hildebrandt@uni-oldenburg.de

Telephone +49 (0)441 798-4629

2: Aging Research Center
NVS Department, Karolinska Institutet and Stockholm University
Tomtebodavägen 18A
17165 Stockholm, Sweden

martin.lovden@ki.se

Telephone +46 (0) 8-690 58 79

benjamin.garzon@ki.se

Appendix D

Prof. Dr. Chris Chambers, Editor
Royal Society Open Science
Cardiff University
CUBRICK, Maindy Road
Cardiff CF24 4HQ

Oldenburg, July 7th, 2019

Dear Professor Chris Chambers, dear reviewer,

We thank you again for granting us the opportunity for this minor revision of our registered report at stage 2. In the following, you will find our replies describing in which ways we addressed the comments and suggestions.

The resubmitted version of the article contains revisions that are highlighted in dark red color.

Associate Editor Comments to Author (Professor Chris Chambers):

The Stage 2 manuscript was returned to one of the original Stage 1 reviewers (the other reviewer was not available), and I also read the manuscript myself. As you will see, the reviewer is overall very positive, which is a testament to the high quality of the study and judicious attention to detail throughout the submission. Nevertheless there are some relatively minor issues to address concerning the reporting and interpretation of results. In revising the manuscript to address the reviewer's comments, please take special care (a) not to change the introduction or framing of the study, and (b) not to allow the outcomes of exploratory post hoc analyses to dominate the interpretation of the results over the outcomes of the preregistered analyses, in either the Discussion or the Abstract. On an additional minor note, please ensure that the description of the Stage 1 manuscript is consistent across the various footnotes, e.g replacing "In the registered manuscript" in Footnote 6 with "In the accepted protocol..."

Thank you for reading the manuscript and providing your opinion. We changed the footnote 6. We also tried to adhere to your suggestion not to focus mainly on the results from the post-hoc analyses. However, we are facing the difficult situation that all of the preregistered hypotheses were falsified. With the non-significant results, we cannot contribute to the understanding of the relationship between CT and cognitive behavior. All we might be able to do when focusing only of the preregistered pipeline is to state that our hypotheses were falsified given that specific pipeline. However, we think that the results of the post-hoc analyses in addition to the preregistered ones are interesting to the community and should be reported and interpreted. There is naturally an imbalance here: the results from the post-hoc analyses are richer and require more interpretation.

Therefore, we now emphasize more strongly in the Discussion and Abstract that the results we can interpret are acquired from exploratory analyses and require cross-validation. We hope you can agree with this view.

Concerning the reviewer's comment: "Although it might not be in line with the policy of preregistration having reader go through text sections that later turn out as irrelevant because the underlying procedure is flawed seems somewhat unnecessary." -- as the reviewer anticipates, it is important to maintain an accurate history of the methodological plan, therefore please do not remove this text.

Comments to Author:

Reviewer: 2

Comments to the Author(s)

Whether the data are able to test the authors' proposed hypotheses by passing the approved outcome-neutral criteria (such as absence of floor and ceiling effects or success of positive controls)

The data are really strong. The decisions the authors made in stage were smart and carefully considered. Nevertheless, analysis of the true data showed that some minor aspects of a priori settings were too strict (i.e. initial analysis with 200 subjects). It is nice, that the authors do not try to hide such issues. The same goes for the correlated errors on the behavioral side. They seem to make sense to me (and should have been prespecified). I cannot easily comment on the meaningfulness of the PM DLPFC correlation amongst residuals.

I am convinced for example that the analysis of competing methods for determining CT deserves a lot of attention in the field and the wisest decision could have been to publish analysis on this problem separately. This is probably not what should happen in the process of publishing a preregistered report. The analysis reported here shows, that using competing procedures for measuring CT delivers vastly different results. Not more than one method is best suited to measure CT. Therefore, for example, earlier results that are based on different methods are likely to deliver wrong results. Indeed, the present paper provides sobering results for supporters of CT-theories.

The surprising and important result, that competing approaches for competing CT deliver vastly different results and are different in reliability should be elaborated. Such a section for example should also include the magnitude of the relation of competing approaches to express CT across subjects.

Thank you for your overall positive view on our work and for recognizing our attempts to be transparent in handling the data. Following your suggestion, we extended the section 6.4 *Reliability of CT Measures in the Extended ROIs* by further information on the relationship between alternative measures of CT as implemented in exploratory analysis above the confirmatory ones.

As mentioned in the respective section of the article, the most direct way to address your comment would have been to provide correlations of latent variables of CT measures in the face network. However, in 5 mm and 10 mm ROIs, these factors were not identified. Therefore, we instead provide a supplementary figure depicting the relationships between measurements at a manifest level, which also clearly delivers the picture showing that the CT results from functional localization and the across-subject masks are quite distinct. The range of the correlations between the different measurement approaches is given in the text. Correlations between CT measures acquired using different functional localizations are much higher than correlations between the mask-based CT values and the results from functional localization. This is not surprising, but important to be emphasized. We now added a few sentences to the discussion elaborating on alternatives to CT measurement.

Some of the important results should be moved from the supplement to the main text. After table 4 the authors should report associations of competing approaches to CT. Figures of CFA/SEM should include standardized parameter estimates or separate tables should be provided.

We agree with you that the standardized parameter estimates are crucial information. However, three out of the four figures depicting measurement/structural models for hypothesis testing were not in the results part, but in the methods part to visualize the hypothesis. Furthermore, if we wanted to include standardized parameters to the figures, two figures per model would be needed in order to represent the slightly divergent parameters for CT in the left and right hemisphere. We fear that this would impair the readability of the article or render the figures rather overloaded. Another option would be to include the tables of the supplements into the article, but the article is already filled densely with information. Again, the readability would be an issue. We thus propose the following solution: We added a range of standardized factor loadings for each latent factor besides the regression weights already mentioned in the text. This way, we hope that the reader gets an idea of the magnitude. For all details, the reader will consult the supplements.

Given only the ROI mask procedure seems to work in the present data it doesn't strike me as essential to also report results from procedures that seemingly fail to achieve what they were designed for. Although it might not be in line with the policy of preregistration having reader go through text sections that later turn out as irrelevant because the underlying procedure is flawed seems somewhat unnecessary.

The Associate Editor already responded to this, confirming the notion that removing the text does not conform to the idea of the registered report.

To be clear – I like the way data are handled here and I think over and beyond expected results the authors should also report upon the correlation of competing measures for measuring CT. I think the discussion should make some key results a little clearer. Agreed – the face factor wasn't very strong, but gathering stronger data together with brain data isn't strongly encouraged by the present data set. And the authors don't make this statement really clear. The same goes for the relation between CT and g. The conclusion drawn from the present data must be that the relation between the two is very small – despite the large sample size and statistical power non adherents of the theory that CT determines g would state it is zero. Given the authors use superior methods and data relative to earlier reports, odds are that earlier reports on a positive association are false positives. The present report should take a clear position concerning the consistency of different data sets.

Thank you for your ideas. We agree that the results do not suggest that behavior is strongly driven by individual differences in CT. Of course the face-specific performance factor is not based on a very rigorous measurement of face cognition. Thus, we argue that it would be very interesting to observe the relationship between a strong face cognition factor and the network CT in the core face network. However, you are right to point out that our study calls the results of smaller samples into question, as the relationship between brain and behavior are very small here. We have altered the discussion with this notion in mind and hope that it is now more clearly delivered.

Whether the Introduction, rationale and stated hypotheses are the same as the approved Stage 1 submission

Yes

Whether the authors adhered precisely to the registered experimental procedures

Sufficiently

Where applicable, whether any unregistered exploratory statistical analyses are justified, methodologically sound, and informative

Yes, absolutely.

Whether the authors' conclusions are justified given the data

Yes.

Thank you again for your insightful comments and helpful ideas. We hope that we were able to address your last concerns in full. Otherwise, please let us know.

Sincerely,

Kristina Meyer, Benjamín Garzón, Martin Lövdén, Andrea Hildebrandt